# PROSER1 mediates TET2 O-GlcNAcylation to regulate DNA demethylation on UTX-dependent enhancers and CpG islands

Xiaokang Wang[1,*], Wojciech Rosikiewicz[2,*], Yurii Sedkov[1,*], Tanner Martinez[1], Baranda S Hansen[1], Patrick Schreiner[2], Jesper Christensen[3], Beisi Xu[2], Shondra M Pruett-Miller[1], Kristian Helin[3,4], Hans-Martin Herz[1]

DNA methylation at enhancers and CpG islands usually leads to gene repression, which is counteracted by DNA demethylation through the TET protein family. However, how TET enzymes are recruited and regulated at these genomic loci is not fully understood. Here, we identify TET2, the glycosyltransferase OGT and a previously undescribed proline and serine rich protein, PROSER1 as interactors of UTX, a component of the enhancer-associated MLL3/4 complexes. We find that PROSER1 mediates the interaction between OGT and TET2, thus promoting TET2 O-GlcNAcylation and protein stability. In addition, PROSER1, UTX, TET1/2, and OGT colocalize on many genomic elements genome-wide. Loss of PROSER1 results in lower enrichment of UTX, TET1/2, and OGT at enhancers and CpG islands, with a concomitant increase in DNA methylation and transcriptional down-regulation of associated target genes and increased DNA hypermethylation encroachment at H3K4me1-predisposed CpG islands. Furthermore, we provide evidence that PROSER1 acts as a more general regulator of OGT activity by controlling O-GlcNAcylation of multiple other chromatin signaling pathways. Taken together, this study describes for the first time a regulator of TET2 O-GlcNAcylation and its implications in mediating DNA demethylation at UTX-dependent enhancers and CpG islands and supports an important role for PROSER1 in regulating the function of various chromatin-associated proteins via OGT-mediated O-GlcNAcylation.

# Introduction

Enhancers are genomic DNA elements which are able to interact with and activate gene promotors irrespective of their genomic location or orientation, often acting over long genomic distances (Banerji et al, 1981, 1983; de Villiers et al, 1982). Many critical developmental genes are under

the regulation of enhancers (Levine, 2010). The histone mark H3K4me1 is highly enriched at active enhancers with significantly lower enrichment at poised/inactive enhancers, whereas H3K4me3 mainly occurs at gene promoters (Heintzman et al, 2007, 2009; Rada-Iglesias et al, 2011). We and others have shown that the epigenetic regulators MLL3 and MLL4 (also known as KMT2C and KMT2D) function as major H3K4 monomethyltransferases on enhancers and are required for enhancer activation during developmental transitions (Herz et al, 2012; Lee et al, 2013; Hu et al, 2013a; Wang et al, 2016a). MLL3 and MLL4 exist in large protein complexes that also contain the H3K27 demethylase UTX (also known as KDM6A) (Agger et al, 2007; Cho et al, 2007; Lee et al, 2007; Mohan et al, 2011; Rickels et al, 2020). Understanding how the MLL3/4 complexes regulate chromatin structure and function to control enhancer activity and transcription is of high importance as UTX, MLL3, and MLL4 belong to some of the most frequently mutated genes across a broad spectrum of adult and pediatric cancers and are also mutated in various neurodevelopmental disorders (Huether et al, 2014; Herz, 2016; Bailey et al, 2018; Priestley et al, 2019; Lavery et al, 2020). Thus, identifying the pathways that regulate the recruitment and function of the MLL3/4 complexes at genomic elements will provide a foundational framework for the development of future therapeutic approaches to foster the treatment of multiple human diseases involving the UTX/MLL3/MLL4 axis.

# Results

### Identification of PROSER1, a novel proline- and serine-rich protein, the DNA demethylase TET2, and the glycosyltransferase OGT as factors that associate with the MLL3/4 complexes

To identify new components that functionally intersect with the MLL3/4 complexes, we purified FLAG-UTX from human embryonic kidney (HEK293) cells. By mass spectrometry (MS) analysis we detected the H3K4 methyltransferases MLL3 and MLL4, and all

[1]Department of Cell and Molecular Biology, St. Jude Children's Research Hospital, Memphis, TN, USA [2]Center for Applied Bioinformatics, St. Jude Children's Research Hospital, Memphis, TN, USA [3]Biotech Research and Innovation Centre and The Novo Nordisk Foundation for Stem Cell Biology, University of Copenhagen, Copenhagen, Denmark [4]Cell Biology Program and Center for Epigenetics Research, Memorial Sloan Kettering Cancer Center, New York City, New York, USA

Correspondence: hans-martin.herz@stjude.org
*Xiaokang Wang, Wojciech Rosikiewicz, and Yurii Sedkov contributed equally to this work

previously reported core subunits and complex-specific subunits of the MLL3/4 complexes (Fig 1A). Furthermore, we also recovered three novel UTX interactors: PROSER1, a protein of unknown function, the methylcytosine dioxygenase TET2 and the glycosyl-transferase OGT (Fig 1A) (He et al, 2011; Ito et al, 2011). The interaction of UTX with PROSER1, TET2 and OGT was further confirmed by Western blotting (WB) (Fig 1B). To assess the relationship between UTX, PROSER1, and TET2 in more detail we FLAG affinity-purified UTX followed by glycerol gradient fractionation and observed that PROSER1 and TET2 co-migrated with UTX in fractions also containing RBBP5, a core subunit of the MLL3/4 complexes (Fig 1C). These

results indicate that PROSER1 and TET2 associate with the MLL3/4 complexes.

To identify a potential function for PROSER1, we purified overexpressed FLAG-PROSER1 from HEK293 cells followed by MS. We detected OGT and all three proteins of the TET family of DNA demethylases as PROSER1 interactors (Fig 2A), which was corroborated independently by WB (Fig S1A). To better mimic native conditions of PROSER1 protein levels, we generated a FLAG-HA-NeonGreen knock-in cell line targeting the N-terminus of PROSER1 (hereafter FHNG-PROSER1) (Fig S1B). FLAG immunoprecipitation (IP) followed by WB from this cell line further confirmed the interaction of PROSER1 with OGT, TET1, and TET2 (Fig 2B). Although UTX

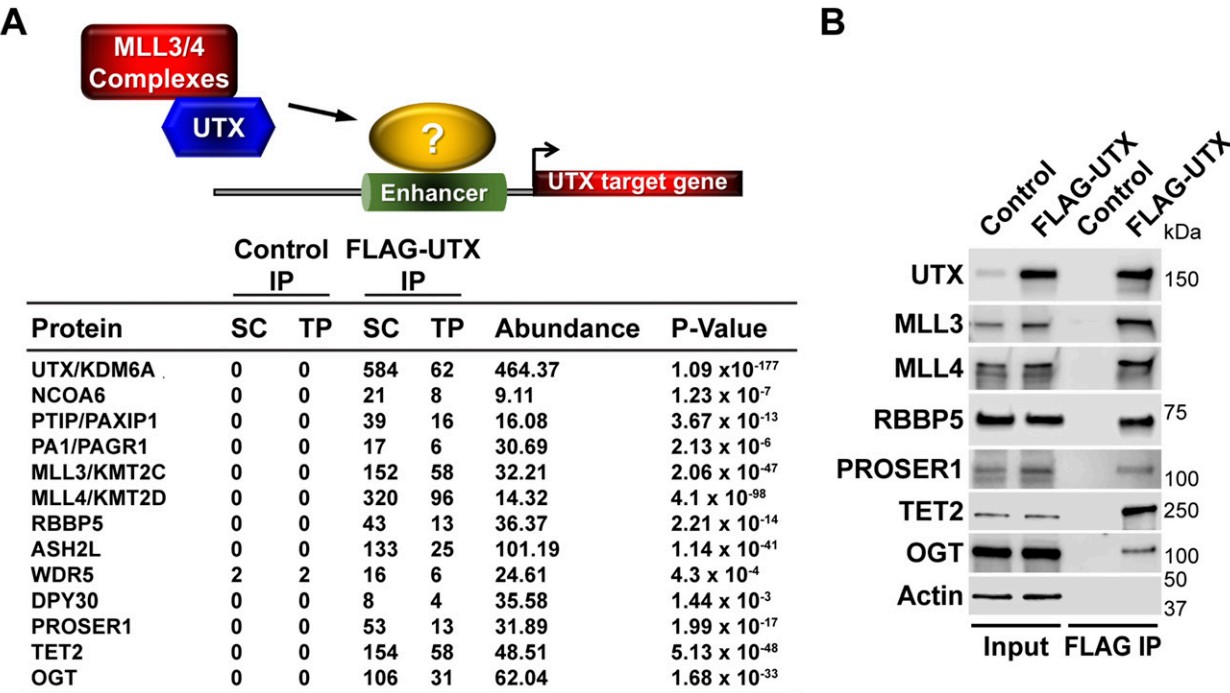

| Protein | Control IP SC | Control IP TP | FLAG-UTX IP SC | FLAG-UTX IP TP | Abundance | P-Value |
|---|---|---|---|---|---|---|
| UTX/KDM6A | 0 | 0 | 584 | 62 | 464.37 | $1.09 \times 10^{-177}$ |
| NCOA6 | 0 | 0 | 21 | 8 | 9.11 | $1.23 \times 10^{-7}$ |
| PTIP/PAXIP1 | 0 | 0 | 39 | 16 | 16.08 | $3.67 \times 10^{-13}$ |
| PA1/PAGR1 | 0 | 0 | 17 | 6 | 30.69 | $2.13 \times 10^{-6}$ |
| MLL3/KMT2C | 0 | 0 | 152 | 58 | 32.21 | $2.06 \times 10^{-47}$ |
| MLL4/KMT2D | 0 | 0 | 320 | 96 | 14.32 | $4.1 \times 10^{-98}$ |
| RBBP5 | 0 | 0 | 43 | 13 | 36.37 | $2.21 \times 10^{-14}$ |
| ASH2L | 0 | 0 | 133 | 25 | 101.19 | $1.14 \times 10^{-41}$ |
| WDR5 | 2 | 2 | 16 | 6 | 24.61 | $4.3 \times 10^{-4}$ |
| DPY30 | 0 | 0 | 8 | 4 | 35.58 | $1.44 \times 10^{-3}$ |
| PROSER1 | 0 | 0 | 53 | 13 | 31.89 | $1.99 \times 10^{-17}$ |
| TET2 | 0 | 0 | 154 | 58 | 48.51 | $5.13 \times 10^{-48}$ |
| OGT | 0 | 0 | 106 | 31 | 62.04 | $1.68 \times 10^{-33}$ |

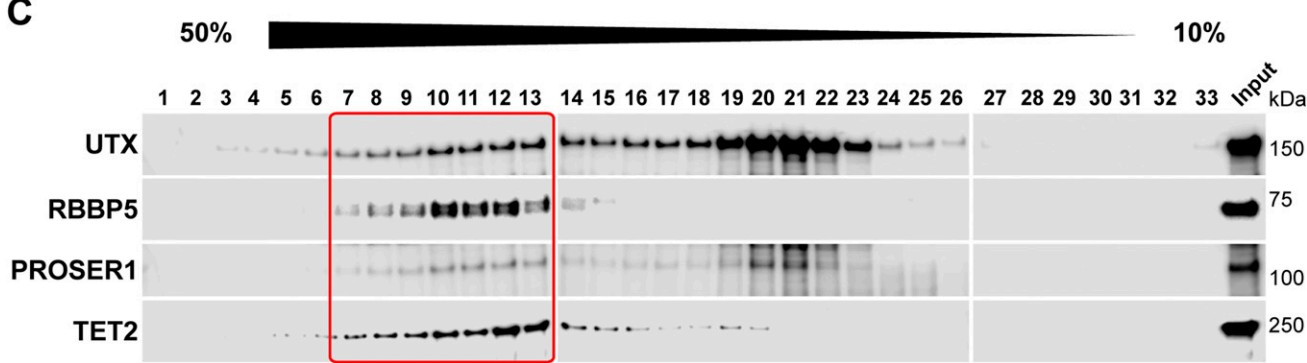

**Figure 1. The MLL3/4 complexes associate with PROSER1, a novel proline and serine rich protein, the DNA demethylase TET2 and the glycosyltransferase OGT.**
**(A)** FLAG-UTX immunoprecipitation (IP) followed by mass spectrometry identifies all known subunits of the MLL3/4 complexes along with PROSER1, a novel serine and proline rich protein, the DNA demethylase TET2 and the glycosyltransferase OGT. SC, spectral counts; TP, peptide counts; abundance = SC × 50 (kD)/protein size (kD). **(B)** Western blot of FLAG-UTX IP from HEK293 cells confirming interaction of UTX with PROSER1, TET2 and OGT. UTX interacts with the H3K4 methyltransferases MLL3 and MLL4, RBBP5 a core component of the MLL3/4 complexes, PROSER1, TET2, and OGT. HEK293 cells with a FLAG-tag expressing plasmid were used as an IP control. Nuclear extracts were used as input. Actin was used as a loading control for the inputs. **(C)** Glycerol gradient sedimentation after FLAG-UTX IP reveals co-fractionation of PROSER1 and TET2 with components of the MLL3/4 complexes (red box).

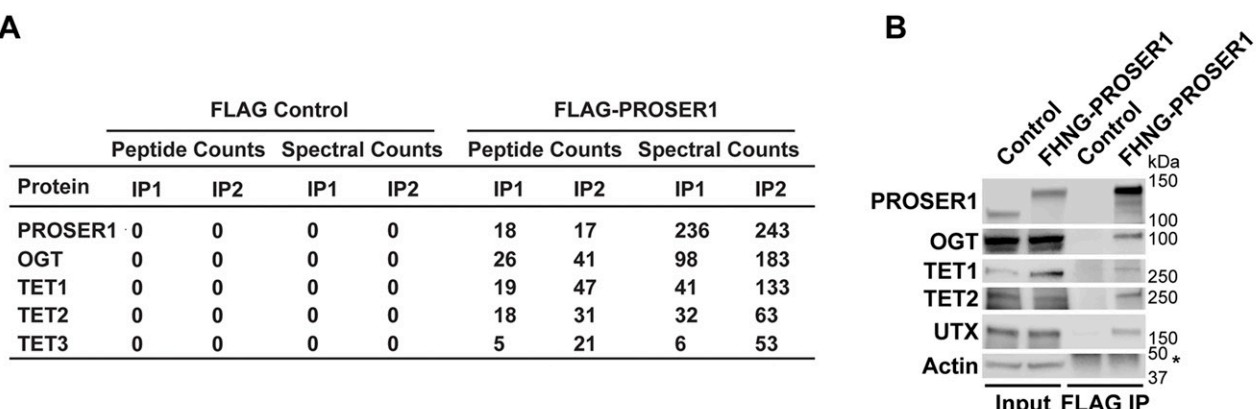

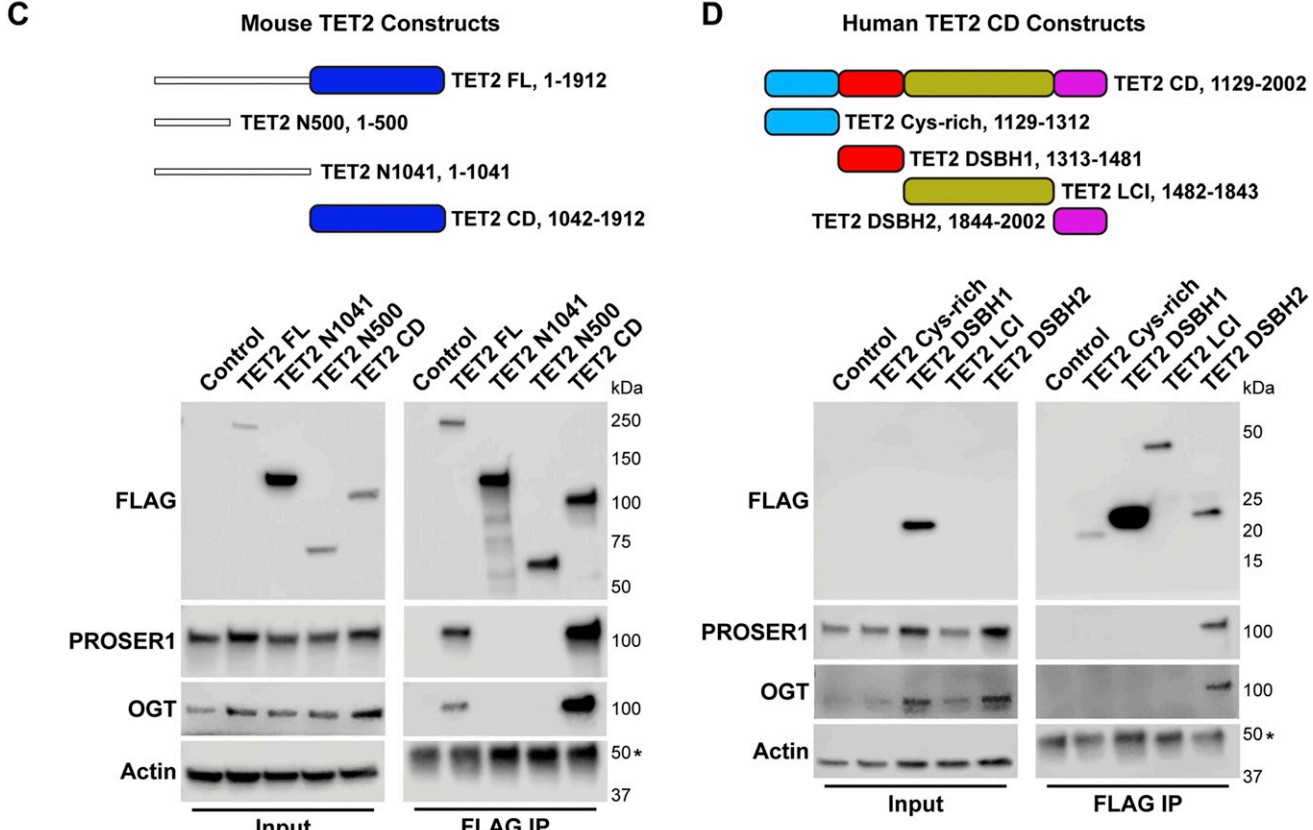

**Figure 2. PROSER1 interacts with members of the TET family of DNA demethylases and OGT.**

**(A)** FLAG-PROSER1 IP followed by mass spectrometry identifies the glycosyltransferase OGT and all three TET family proteins. **(B)** Western blot of FLAG IP from *FLAG-HA-NeonGreen-PROSER1* (*FHNG-PROSER1*) knock-in HEK293 cells confirming interaction of PROSER1 with OGT, TET1, TET2, and UTX. Wild-type (WT) HEK293 cells were used as an IP control. Nuclear extracts were used as input. Actin was used as a loading control for the inputs. The asterisk indicates the IgG heavy chain. **(C)** Top: Domain structure and constructs of mouse TET2. Blue: cysteine-rich dioxygenase (CD) domain. TET2 FL, full length construct. TET2 N500, N-terminal 500 amino acids (aa). TET2 N1041, N-terminal 1,041 aa. TET2 CD, TET2 cysteine–rich dioxygenase domain. Bottom: HEK293 cells were transiently transfected with a series of FLAG-tagged mouse TET2 constructs as shown on the top. WB of FLAG IPs from total cell lysates showing interaction of PROSER1 and OGT with TET2 FL and TET2 CD. Total cell lysates were used as inputs. Actin was used as a loading control for the inputs. The asterisk indicates the IgG heavy chain. **(D)** Top: Domain structure and constructs of the human TET2 cysteine–rich dioxygenase (CD) domain. Light blue, cysteine-rich domain. Red, double-stranded β-helix (DSBH1) domain 1. Olive, low-complexity insert (LCI) region. Purple, double-stranded β-helix (DSBH2) domain 2. Bottom: HEK293 cells were transiently transfected with a series of FLAG-tagged human TET2 constructs as shown on the top. WB of FLAG IPs from total cell lysates depicting interaction of PROSER1 and OGT with TET2 DSHB2. Total cell lysates were used as inputs. Actin was used as a loading control for the inputs. The asterisk indicates the IgG heavy chain. TET2 Cys-rich, TET2 LCI, and TET2 DSHB2 could not be detected in the inputs.

could not be identified by MS after FLAG-PROSER1 IP (Fig 2A), we were able to confirm the interaction between PROSER1 and UTX by WB from the FHNG-PROSER1 and FLAG-PROSER1 IPs (Figs 2B and S1A). As UTX and PROSER1 both pulled down TET2, we next mainly focused on TET2.

## PROSER1 mediates TET2 O-GlcNAcylation and stability by promoting the interaction between OGT and TET2

To map the TET2 region(s) that interact(s) with PROSER1, we leveraged a series of FLAG-tagged TET2 constructs: TET2 FL, TET2 N500, TET2 N1041, and TET2 CD (Nakagawa et al, 2015) (Fig 2C). After transfection of these constructs into HEK293 cells followed by FLAG IP, we found that only TET2 FL and TET2 CD pulled down PROSER1 and OGT (Fig 2C). Similarly, transfection of these constructs into FHNG-PROSER1 cells followed by HA IP only identified TET2 FL and TET2 CD as PROSER1 interactors (Fig S1C). The interaction between TET2 and OGT and more specifically TET2 CD and OGT has been reported previously (Chen et al, 2013; Deplus et al, 2013; Vella et al, 2013). To further dissect the interaction between TET2 CD and PROSER1, we divided TET2 CD into four smaller domains: TET2 Cys-rich, TET2 DSBH1, TET2 LCI, and TET2 DSBH2 (Fig 2D). Our results show that PROSER1 and OGT only associated with TET2 DSBH2 (Fig 2D). As OGT was shown to bind a conserved C-terminal region within the DSBH2 domain of TET1 (Hrit et al, 2018), we further dissected TET2 DSBH2 into a 92-amino acid (aa) N-terminal and a 66 aa C-terminal region according to Hu et al (2013b) (Fig 3A). Interestingly, we found that only the C-terminal region of TET2 DSBH2 could interact with both PROSER1 and OGT (Fig 3A). The intimate relationship between OGT and PROSER1 motivated us to test, if PROSER1 might mediate the interaction between OGT and TET2. For this purpose, we used CRISPR/Cas9 editing to generate PROSER1 KOs in HEK293 cells (Fig S1D). By IP of overexpressed TET2 DSBH2 in WT and PROSER1 KO cells, we found that significantly less OGT was immuno-precipitated by TET2 DSBH2 in PROSER1 KO cells compared with WT cells even though more TET2 DSBH2 was immunoprecipitated in PROSER1 KO cells (Fig 3B). At the same time OGT protein levels were unchanged between WT and PROSER1 KO cells (Fig 3B–D). Furthermore, after IP of endogenous TET2, with nearly equal TET2 protein amounts immuno-precipitated from WT and PROSER1 KO cells, we observed that less OGT was co-immunoprecipitated from PROSER1 KO compared with WT cells (Fig 3C). As OGT has been shown to glycosylate TET proteins by O-GlcNAcylation (Bauer et al, 2015; Hrit et al, 2018), we probed the O-GlcNAcylation level of TET2 immunoprecipitated from WT and PROSER1 KO cells and found that O-GlcNAcylated TET2 could no longer be detected in PROSER1 KO compared with WT cells (Fig 3C). As O-GlcNAcylation has been reported to increase protein stability (Sola & Griebenow, 2009; Yang & Qian, 2017; Konzman et al, 2020), we wanted to test whether TET1 and TET2 protein levels were altered in PROSER1 KO cells. Indeed, both TET1 and TET2 protein levels were decreased, whereas TET1 and TET2 mRNA levels were unchanged in PROSER1 KO compared with WT cells (Figs 3D, S1D, and S2). However, OGT protein levels remained unchanged (Fig 3D). In addition, siRNA-mediated knockdown of OGT resulted in reduced PROSER1 protein levels and O-GlcNAcylated PROSER1, suggesting that OGT directly O-GlcNAcylates PROSER1 and thus regulates its protein stability (Fig 3E). Overall, our data suggest that OGT directly interacts with and O-GlcNAcylates both PROSER1 and TET2, thus regulating their protein stability. and that PROSER1 mediates the re-cruitment of OGT to TET2 (Fig 3F).

## PROSER1 regulates the chromatin association of TET1/2 to mediate UTX/H3K4me1-dependent enhancer activity

To obtain a genome-wide overview of the relationship between PROSER1, UTX, OGT, TET1 and TET2, we performed ChIP-seq for these epigenetic factors and the histone marks H3K4me1, H3K4me2, H3K4me3, and H3K27ac. Because of the lack of reliable commercial PROSER1 antibodies, we generated antibodies against three different PROSER1 antigens to perform PROSER1 ChIP-seq in WT and PROSER1 KO cells. In total, 18,017 reproducible PROSER1 peaks were identified (Fig 4A and B). These 18,017 PROSER1 peaks were also co-occupied by UTX, OGT, TET1, and TET2 in WT cells (Fig 4A and B). When centered on UTX, TET1, TET2, or OGT peaks a similar co-occupancy with all other epigenetic factors was observed (Fig S3A–D). Whereas most PROSER1, TET2, and OGT peaks were detected within pro-moter regions, a large proportion of UTX and TET1 peaks were also located within introns in addition to promoter regions (Fig S3E). Almost all PROSER1 peaks (99%) overlapped with UTX and 63% of PROSER1 peaks co-localized with TET2 (Fig 4B). To test whether PROSER1 is required for the genome-wide occupancy of the MLL3/4 complexes (as assessed by UTX occupancy), TET1, TET2, and OGT, we also performed ChIP-seq of the abovementioned epi-genetic factors and histone marks in PROSER1 KO cells. In total, we identified 27,300 regions with lower UTX and 14,342 regions with lower H3K4me1 enrichment in PROSER1 KO versus WT cells (FC ≥ 2), among which 4,421 regions displayed lower combined UTX and H3K4me1 enrichment (Fig S3F). These 4,421 regions also showed reduced enrichment of TET1, TET2, OGT, H3K4me2, and H3K27ac (Fig 4C). Interestingly, most of these 4,421 loci lack H3K4me3 and are distal from promoters, suggesting that they mainly constitute enhancers (Figs 4C and S3G). Fig 4D shows individual examples of these 4,421 regions with lower UTX and H3K4me1 enrichment where loss of PROSER1 also causes a reduction of TET1, TET2, OGT, H3K4me2, and H3K27ac at the DDB2 promoter and SOX2 enhancer resulting in transcriptional down-regulation of DDB2 and SOX2 (Fig 4D and E). Importantly, the reduction of UTX, TET1, TET2, and OGT at these 4,421 loci in PROSER1 KO cells is also associated with the repression of their nearest target gene (Fig 4F). In summary, this indicates that PROSER1 controls the recruitment of TET1/2 and OGT to regulate UTX-dependent enhancer activity.

## PROSER1 regulates DNA demethylation at UTX/H3K4me1-dependent enhancers and CpG islands

TET enzymes catalyze the successive oxidation of 5-methylcytosine (5mC) to 5-hydroxymethylcytosine (5hmC), 5-formylcytosine (5fC), and 5-car-boxylcytosine (5caC) (Tahiliani et al, 2009; He et al, 2011; Ito et al, 2011). Based on our findings that PROSER1 regulates TET1 and TET2 protein stability and their occupancy on UTX/H3K4me1-dependent enhancers (Figs 3D and 4C and D), we thus investigated how the genome-wide DNA methylation pattern was affected in PROSER1 KO cells. Whereas no global change in 5mC could be detected, 5hmC, 5fC, and 5caC all were signif-icantly reduced in PROSER1 KO compared with WT cells (Fig 5A). After carrying out whole genome bisulfite sequencing (WGBS), we ob-served approximately equal numbers of hypermethylated and hypomethylated cytosines of which both categories displayed a similar genomic distribution pattern (Fig S4A and B). However, the proportion of hypermethylated CpG islands (CGIs) and promoters was significantly increased over hypomethylated CGIs and pro-moters in PROSER1 KO versus WT cells (Fig S4A). At the 4,421 UTX/H3K4me1-dependent regions, DNA methylation was increased,

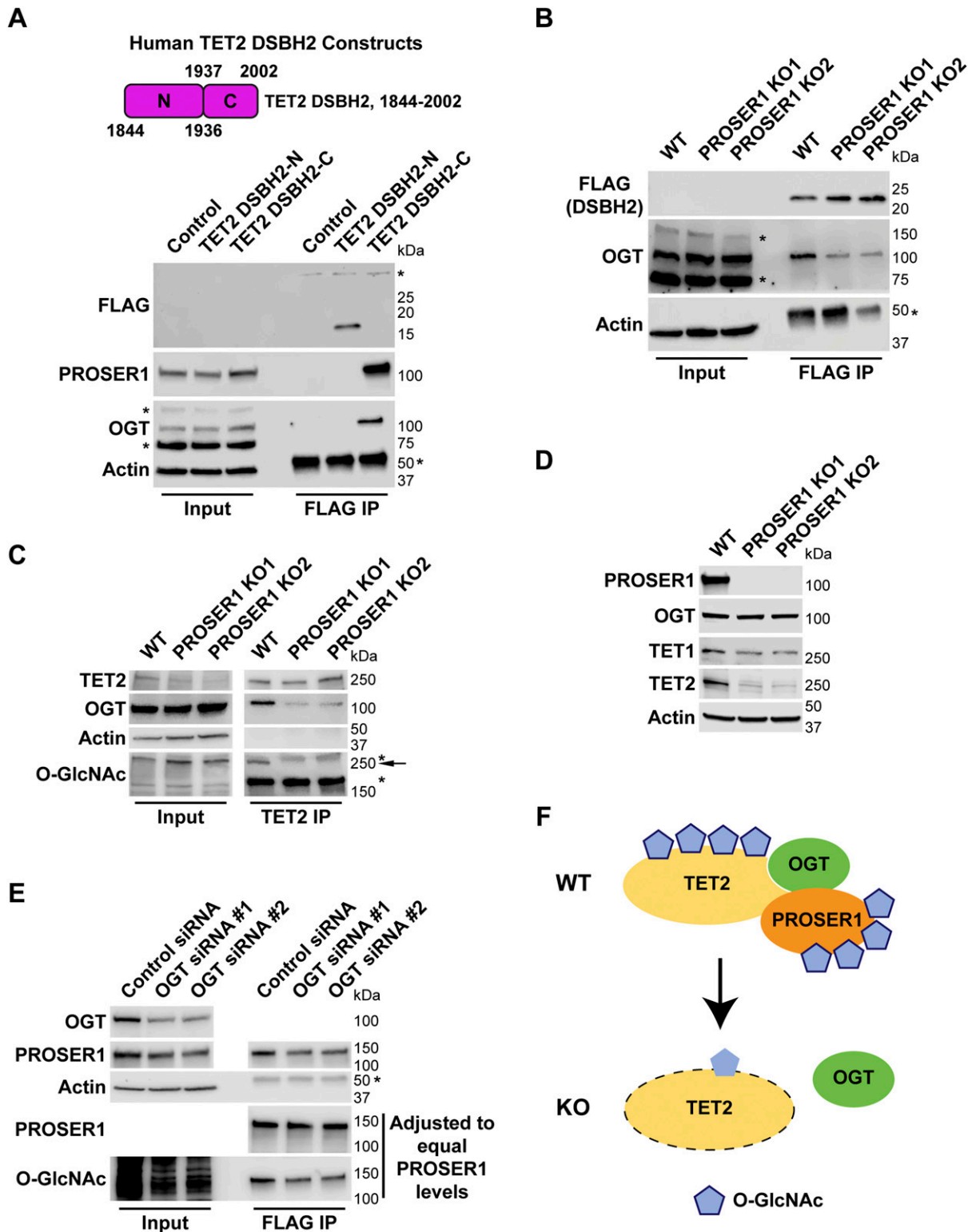

**Figure 3. PROSER1 mediates TET2 O-GlcNAcylation and stability by promoting the interaction between OGT and TET2.**

**(A)** Top: Domain structure and constructs of the human TET2 double-stranded β-helix 2 (DSBH2) domain. TET2 DSHB2-N: N-terminal 93 aa of DSHB2. TET2 DSHB2-C: C-terminal 66 aa of DSHB2. Bottom: HEK293 cells were transiently transfected with the FLAG-tagged constructs as shown on the top. Western blot of FLAG IPs showing interaction of PROSER1 and OGT with TET2 DSHB2-C. Total cell lysates were used as inputs. Actin was used as a loading control for the inputs. The asterisks indicate an unspecific band (upper panel) or the IgG heavy chain (lower panel). TET2 DSHB2-N and TET2 DSHB2-C could not be detected in the inputs. Failure to detect FLAG-TET2 DSHB2-C after FLAG IP might be due to its positive charge at the pH of the sample buffer. **(B)** WT and *PROSER1* KO HEK293 cells were transiently transfected with FLAG-tagged TET2

whereas 5hmC DNA immunoprecipitation followed by sequencing (hMeDIP-seq) revealed strongly reduced 5hmC levels in *PROSER1* KO versus WT cells (Fig 5B), suggesting compromised TET activity at these loci. At the identified 128 hypermethylated CGIs, 5hmC levels were also reduced in *PROSER1* KO versus WT cells (Fig 5C). Likewise, UTX, TET1, TET2, OGT, H3K4me1, H3K4me2, and H3K4me3 enrichment was also dampened at these CGIs in *PROSER1* KO compared with WT cells (Fig S4C). Accordingly, hypermethylation of these CGIs was associated with gene repression of the nearest associated gene (Fig S5A). Similarly, differentially methylated regions extending over 500 bp (DMR500) or 1,000 bp (DMR1000) which were associated with increased DNA methylation and overlapping with at least one PROSER1, UTX, or H3K4me1 peak showed a decrease of UTX, TET1, TET2, OGT, H3K4me1, H3K4me2, and H3K4me3 enrichment and were also associated with gene repression of their nearest target gene in *PROSER1* KO versus WT cells (Figs 5D, S4D, and S5B and C). To further confirm that 5hmC is reduced globally as observed by dot blot analysis (Fig 5A) and not only restricted to specific sites in *PROSER1* KO versus WT cells we also used the 5hmC measurements from our hMeDIP-seq data. This analysis revealed that *PROSER1* KO compared with WT cells exhibit a strong global decrease on nearly all regions where 5hmC can be detected (Fig S5D).

### Loss of PROSER1 function causes DNA hypermethylation encroachment at CpG islands

Recent research has shown that a notable number of CGI borders predisposed by H3K4me1 may undergo DNA hypermethylation encroachment in cancers (Skvortsova et al, 2019). As PROSER1 and TET2 are associated with UTX, a complex-specific subunit of the MLL3/4 complexes which catalyze H3K4me1 on enhancers (Fig 1A) and some CGIs undergo DNA hypermethylation in *PROSER1* KO cells (Fig 5C), we thus further investigated, if loss of PROSER1 would result in DNA hypermethylation encroachment at CGIs. Indeed, we observe CGIs which either display bidirectional (*IGFBP7*) or monodirectional DNA hypermethylation encroachment from their 5′ (*MGA*) or 3′ (*HOXC6*) ends resulting in decreased transcription of their associated target genes (Figs 6A and S6A). Altogether, we identified 483 CGIs which show bidirectional or monodirectional DNA hypermethylation encroachment in *PROSER1* KO versus WT cells (Fig 6B and C and Table S1). In accordance with these findings, 5hmC levels are strongly reduced at these sites in *PROSER1* KO compared with WT cells, suggesting that these CGIs undergo active PROSER1-mediated DNA demethylation via TET proteins (Fig S6B).

## Discussion

In conclusion, our study identifies PROSER1, TET2 and OGT as novel factors that associate with the MLL3/4 complexes via their complex-specific subunit UTX (Fig 7A). We reveal for the first time a function for the previously undescribed protein PROSER1 in regulating TET2 O-GlcNAcylation and stability by promoting the interaction between TET2 and OGT (Figs 3F and 7A). Furthermore, our identification of all three TET family proteins as PROSER1 interactors suggests a common regulatory mechanism for OGT-mediated O-GlcNAcylation of TET1-3 by PROSER1. However, this remains to be established in further detail for TET1 and TET3 in the future. Genome-wide we find that PROSER1 regulates the chromatin association of TET1/2 to mediate DNA demethylation at UTX/H3K4me1-dependent enhancers and CGIs (Fig 7A). Based on the provided biochemical evidence we propose that OGT-mediated O-GlcNAcylation stabilizes TET1/2 at and/or enhances TET1/2 recruitment to these sites. The dependency of UTX on TET1/2 and thus the regulation of H3K4me1 via the MLL3/4 complexes at these sites could directly rely on the interaction of UTX with TET1/2, result from the lower DNA methylation levels established by the presence of TET1/2, or a combination thereof. Interestingly, loss of PROSER1 also recapitulates the DNA hypermethylation encroachment that occurs at H3K4me1-predisposed CGI borders in several cancers (Skvortsova et al, 2019). Under normal conditions, PROSER1 might carry out a tumor-protective role by clearing DNA methylation from these CGIs through its regulation of TET1/2. Intriguingly, frequent mutations of PROSER1 along with UTX, MLL3, and MLL4 have been described in esophageal squamous cell carcinoma (Gao et al, 2014), indicating that PROSER1/OGT/TET2 and the MLL3/4 complexes might converge in their tumor suppressive functions to protect certain CGIs and/or genomic elements such as enhancers from DNA methylation and thus silencing of associated genes.

We currently cannot rule out the possibility that PROSER1 might also directly inhibit the recruitment or activity of members of the DNA methyltransferase (DNMT) family at UTX/H3K4me1-down-regulated regions, CGIs or hypermethylated DMRs. However, our data show that PROSER1 prominently interacts with all three members of the TET protein family and is required to stabilize TET2 protein levels via OGT-mediated O-GlcNAcylation (Figs 2 and 3). Our model is further corroborated by the strong reduction of TET1, TET2, and of 5hmC, which we observe at CGIs in *PROSER1* KO compared with WT cells (Figs S4C and S6B). In contrast, we were not able to detect an association of any member of the DNA methyltransferase (DNMT) family with PROSER1 by MS and neither were the protein levels of DNMT1, DNMT3A, and DNMT3B affected in *PROSER1* KO compared with WT cells (Fig S7A and data not shown). Therefore, it is

---

DSHB2. WB of FLAG IPs depicts decreased interaction of OGT with TET2 DSHB2 in the absence of PROSER1. Total cell lysates were used as inputs. Actin was used as a loading control for the inputs. The asterisks indicate unspecific bands (middle panel) or the IgG heavy chain (lower panel). TET2 DSHB2 could not be detected in the inputs. **(C)** TET2 IP from nuclear extracts of WT and *PROSER1* KO HEK293 cells. Approximately equal amounts of TET2 were IPed from WT and *PROSER1* KO cells. A significant decrease in OGT binding to TET2 and a strong reduction of TET2 O-GlcNAcylation is observed in *PROSER1* KO versus WT HEK293 cells. Nuclear extracts were used as inputs. Actin was used as a loading control for the inputs. The asterisks indicate unspecific bands, the arrow O-GlcNAcylated TET2. **(D)** WB for the indicated proteins from nuclear extracts of WT and *PROSER1* KO HEK293 cells. Actin was used as a loading control. **(E)** FLAG IP from total cell lysates of *FHNG-PROSER1* HEK293 cells treated with control or *OGT* siRNA. PROSER1 protein levels are decreased upon OGT knockdown. IPed PROSER1 from control and *OGT* siRNA-treated cells was adjusted to equal amounts to show that OGT catalyzes O-GlcNAcylation of PROSER1. Total cell lysates were used as inputs. Actin was used as a loading control for the inputs. The asterisk indicates the IgG heavy chain. **(F)** Model of PROSER1 function. PROSER1 mediates the recruitment of OGT to TET2 to facilitate O-GlcNAcylation of TET2 and PROSER1. In the absence of PROSER1, OGT association with TET2 is impaired resulting in decreased TET2 and PROSER1 O-GlcNAcylation and protein stability.

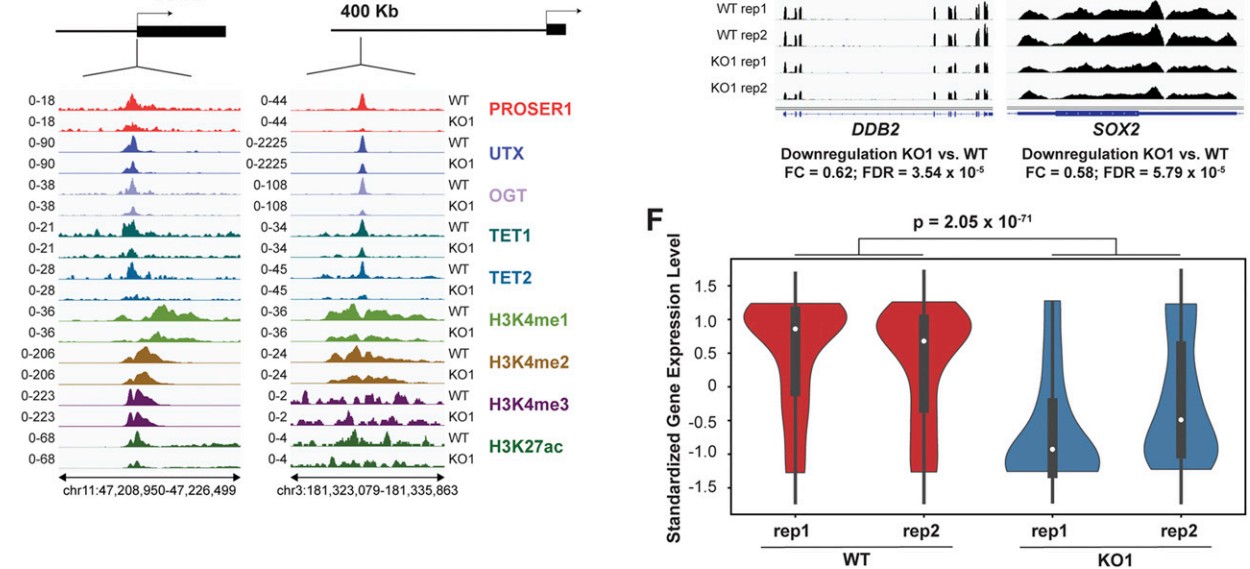

**Figure 4. PROSER1 regulates the chromatin association of TET1/2 to mediate UTX/H3K4me1–dependent enhancer activity.**
**(A)** Heatmaps centered on 18,017 reproducible PROSER1 binding sites obtained from three different PROSER1 antibodies in WT HEK293 cells. Occupancy of PROSER1, UTX, OGT, TET1, TET2, H3K4me1, H3K4me2, H3K4me3, and H3K27ac is displayed. **(B)** Venn diagram depicting the overlap between reproducible PROSER1, UTX, and TET2 peaks in WT HEK293 cells. **(C)** Heatmaps centered on 4,421 UTX/H3K4me1-down-regulated regions in *PROSER1* KO compared with WT cells. Occupancy of PROSER1, UTX, OGT, TET1, TET2, H3K4me1, H3K4me2, H3K4me3, and H3K27ac is displayed. **(D)** Genome browser tracks depicting the ChIP-seq profiles of PROSER1, UTX, OGT, TET1, TET2, H3K4me1, H3K4me2, H3K4me3, and H3K27ac in WT and *PROSER1* KO HEK293 cells at the *DDB2* promoter (left) and *SOX2* enhancer (right). **(E)** Genome browser tracks showing the RNA-seq profiles of the *DDB2* and *SOX2* genes in WT and *PROSER1* KO HEK293 cells. Two replicates are displayed for each genotype. **(F)** Violin plots showing transcriptional down-regulation of the 400 genes associated with the 4,421 regions displaying lower enrichment of UTX and H3K4me1 in *PROSER1* KO compared with WT cells. Expression values were standardized for each gene across the samples. The median interquartile range and the 1.5× interquartile range are shown for each violin plot.

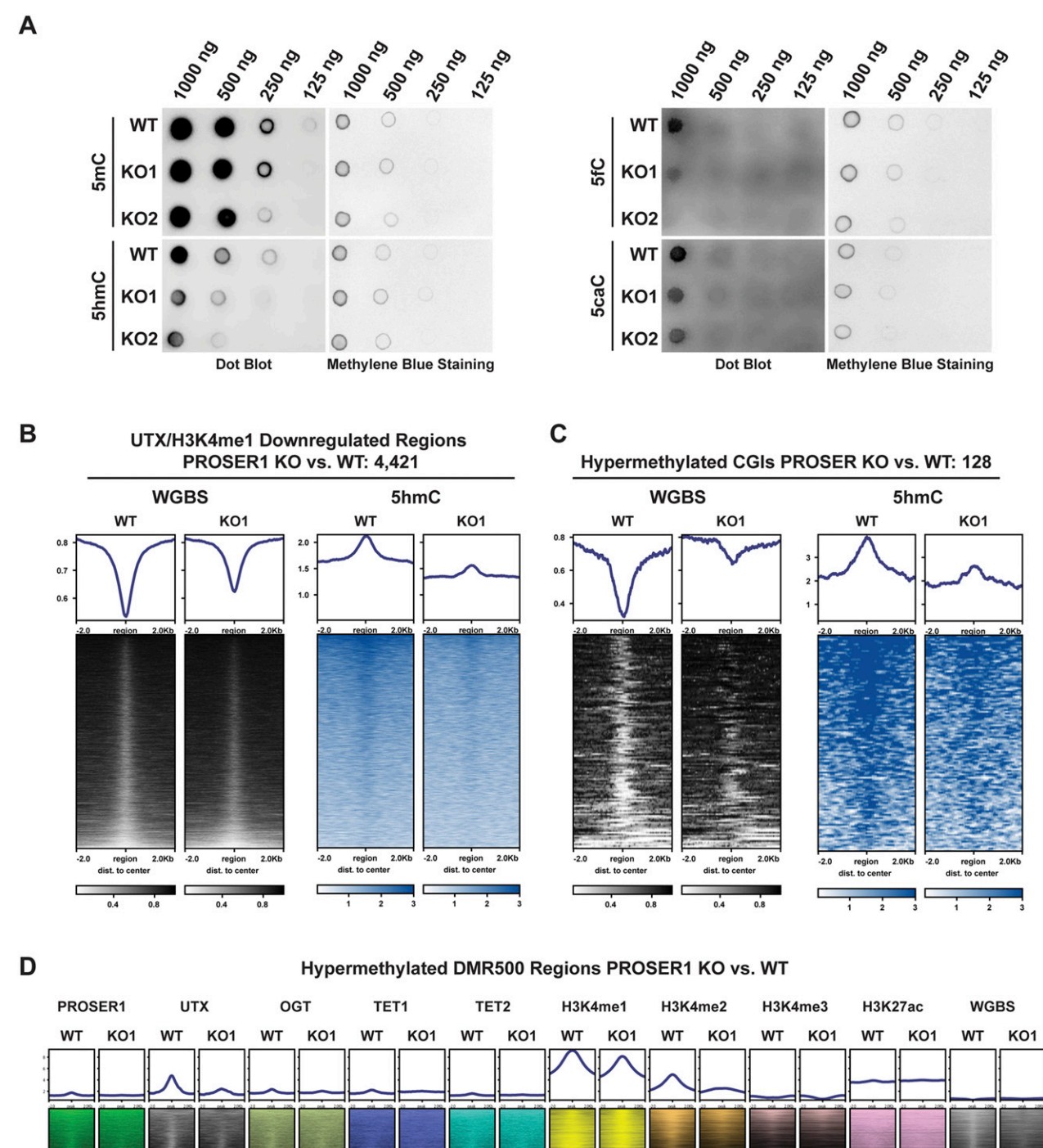

**Figure 5. PROSER1 regulates DNA demethylation on UTX/H3K4me1-dependent enhancers and CpG islands.**
**(A)** Dot blot analysis of genomic 5mC, 5hmC, 5fC, and 5caC in WT and *PROSER1* KO HEK293 cells. 5mC is unchanged and 5hmC, 5fC, and 5caC are decreased in *PROSER1* KO versus WT cells. Methylene Blue staining was performed to ensure that equal amounts of DNA were used from WT and *PROSER1* KO cells. **(B)** Heat maps displaying whole genome bisulfite sequencing (WGBS) and hMeDIP-seq results centered on 4,421 UTX/H3K4me1-down-regulated regions in *PROSER1* KO versus WT cells. **(C)** Heat maps showing WGBS and hMeDIP-seq results centered on 128 hypermethylated CpG islands in *PROSER1* KO compared with WT cells. **(D)** Heat maps centered on hypermethylated DMR500 regions in *PROSER1* KO compared with WT cells overlapping with at least one PROSER1, UTX, or H3K4me1 peak in WT cells. Occupancy of PROSER1, UTX, OGT, TET1, TET2, H3K4me1, H3K4me2, H3K4me3, and H3K27ac and WGBS results are shown.

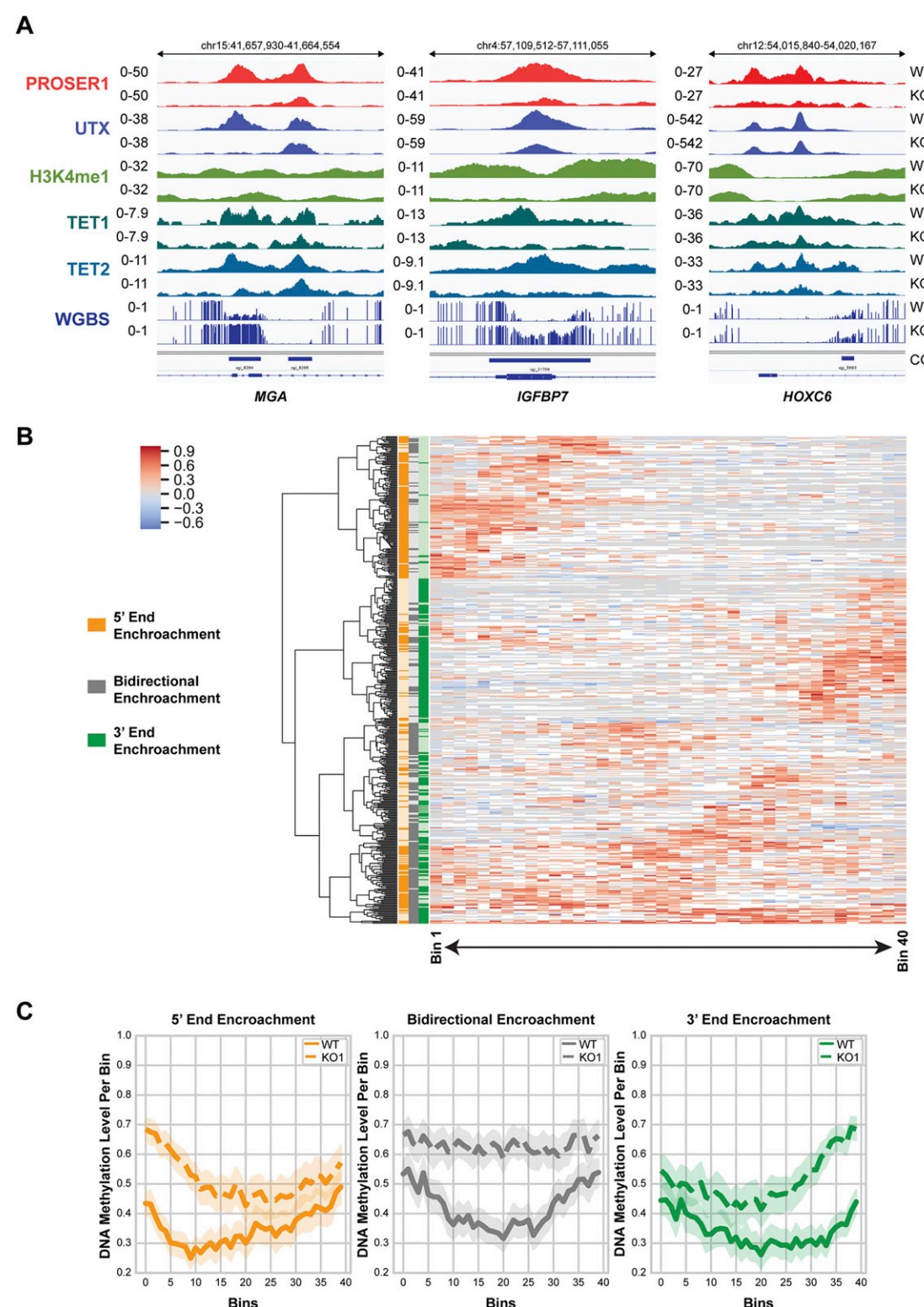

**Figure 6. Loss of PROSER1 function causes DNA hypermethylation encroachment at CpG islands.**
**(A)** Representative genome browser tracks depicting the ChIP-seq profiles of PROSER1, UTX, H3K4me1, TET1, TET2, and whole genome bisulfite sequencing results at CGIs with DNA hypermethylation encroachment in *PROSER1* KO versus WT cells. Examples for 5' encroachment (*MGA*), bidirectional encroachment (*IGFBP7*), and 3' encroachment (*HOXC6*) in WT and *PROSER1* KO cells are displayed. **(B)** Heat map showing the whole genome bisulfite sequencing results of 483 identified CGIs with DNA hypermethylation encroachment in *PROSER1* KO compared with WT cells. Clustering was performed based on the pattern of DNA hypermethylation encroachment: 5' end encroachment (orange), bidirectional encroachment (gray), and 3' encroachment (green). The color scale shows the DNA methylation range difference between *PROSER1* KO and WT cells. Missing values (Not-a-Number; NaN) are depicted in white. **(C)** Average per-bin DNA methylation level of the CGIs with DNA hypermethylation encroachment described in Fig 6B categorized into groups of 5' end encroachment (orange), bidirectional encroachment (gray), and 3' end encroachment (green).

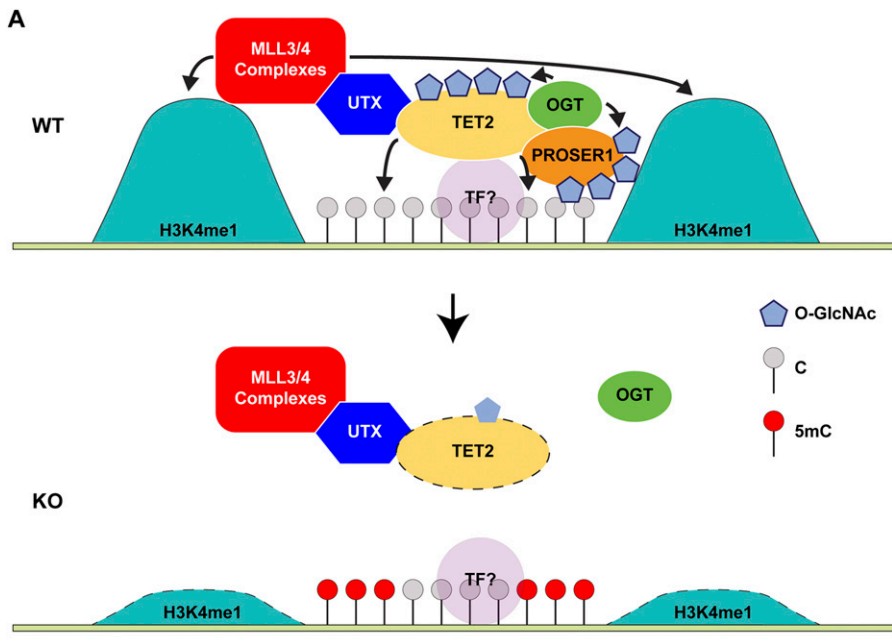

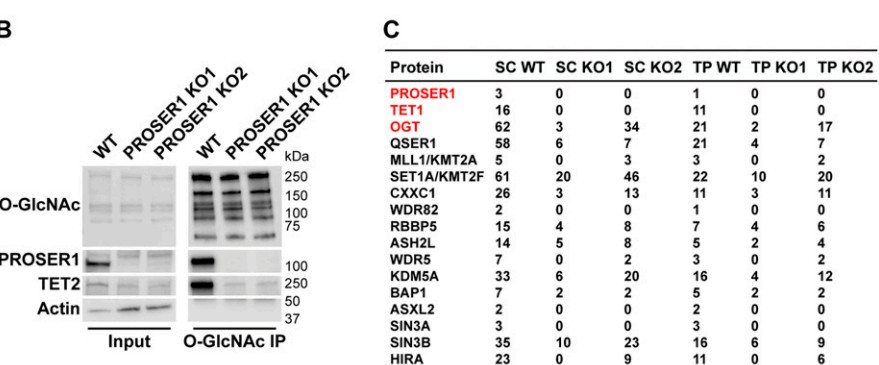

**Figure 7. Model of PROSER1 function at enhancers and CpG islands and broader implication for PROSER1 as a general mediator of OGT activity in chromatin signaling.**
**(A)** Model of PROSER1 function at enhancers and CpG islands. PROSER1 mediates OGT interaction with and O-GlcNAcylation of TET2 to control TET2 stabilization at enhancers and CGIs that are controlled by the activity of the MLL3/4 complexes. TET2 is involved in the recruitment of the MLL3/4 complexes via their complex-specific subunit UTX. In the absence of PROSER1 TET2 stabilization at enhancers and CGIs is impaired resulting in reduced recruitment of the MLL3/4 complexes. TF, transcription factor. **(B)** Western blot of IP with an O-GlcNAc–specific antibody from WT and *PROSER1* KO HEK293 cells confirming strongly reduced O-GlcNAcylation of TET2 in *PROSER1* KO compared with WT cells. Nuclear extracts were used as input. Actin was used as a loading control for the inputs. **(C)** IP with an O-GlcNAc–specific antibody followed by mass spectrometry identifies TET1 and OGT as differentially O-GlcNAcylated in WT versus *PROSER1* KO HEK293 cells. Many components of the H3K4 methyltransferase and demethylase family also display reduced O-GlcNAcylation in *PROSER1* KO compared with WT cells. Of note, QSER1, a protein just recently described as a factor that protects DNA methylation valleys from de novo methylation also showed reduced O-GlcNAcylation in *PROSER1* KO versus WT cells.

unlikely that PROSER1 acts as a direct inhibitor of the DNMT family at the investigated genomic regions in this study. However, based on previous findings by others (Charlton et al, 2020) we anticipate that TET and DNMT family members are operating in a dynamic competitive relationship at overlapping target regions including the UTX/H3K4me1-down-regulated regions, CGIs or hypermethylated DMRs that depend on PROSER1 function. Thus, based on the interdependency between the DNA demethylation and methylation machineries it is to be expected that reduced recruitment of TET proteins at PROSER1-dependent genomic regions will indirectly result in increased recruitment and/or activity of DNMT family members in *PROSER1* KO cells.

Interestingly, we detected approximately equal numbers of hypermethylated and hypomethylated cytosines in *PROSER1* KO compared with WT cells (Fig S4A). Although this result appears to be puzzling for a factor such as PROSER1 that is directly involved in regulating DNA demethylation via the TET protein family, this phenomenon has repeatedly been observed before in cells with an impaired DNA demethylation machinery in both mammals and plants (Hon et al, 2014;

Lu et al, 2014; Wang et al, 2016b, 2019). The decrease in DNA methylation at certain loci in *PROSER1* KO cells might be attributed to the competitive relationship between the DNA methylation and demethylation machineries as mentioned above (Charlton et al, 2020). Alternatively, loss of PROSER1 function may also result in redistribution of members of the TET or DNMT family of proteins. For example, a recent study reported that the loss of TET1 in mESCs resulted in the relocalization of DNMT3A thus providing a potential mechanism for the observed hypomethylated regions in *PROSER1* KO cells (Lopez-Moyado et al, 2019). We found that regions with increased TET1 or TET2 occupancy tend to be hypomethylated, whereas regions with decreased TET1 or TET2 occupancy have the tendency to be hypermethylated in *PROSER1* KO versus WT cells (Fig S7B). However, we also identified hypomethylated regions with decreased TET1 or TET2 occupancy and hypermethylated regions with increased TET1 or TET2 occupancy (Fig S7B). Importantly, both UTX/H3K4me1-down-regulated regions and PROSER1 occupied regions were strongly and dominantly associated with DNA hypermethylation in *PROSER1* KO compared with WT cells (Fig S7B). Overall, this indicates that DNA hypomethylation at some regions may result from increased

TET enrichment because of relocalization of TET proteins or as mentioned above that the increase in TET enrichment on hypomethylated regions might be the consequence of relocalized DNMT proteins in *PROSER1* KO cells.

It is also tempting to speculate that PROSER1 might function as a more general factor in regulating O-GlcNAcylation of other OGT substrates beyond the TET protein family. Interestingly, whereas IPs with an antibody recognizing O-GlcNAcylated proteins followed by MS confirmed a reduction of TET1, TET2, and OGT O-GlcNAcylation in the absence of PROSER1, many other chromatin-associated factors including many components of the H3K4 methyltransferase and demethylase family also displayed reduced O-GlcNAcylation in *PROSER1* KO compared with WT cells (Fig 7B and C). This suggests that PROSER1 might constitute a more general regulator of chromatin signaling pathways that depend on the catalytic activity of OGT. Intriguingly, we also identified QSER1 as a potential PROSER1-dependent OGT substrate (Fig 7C). QSER1 was just recently described as a factor that protects DNA methylation valleys from de novo methylation in association with TET1 (Dixon et al, 2021). But the molecular mechanism by which QSER1 functions remains enigmatic. Our findings indicate that QSER1 and PROSER1 might operate in an analogous manner as part of a collaborative protein network to regulate DNA demethylation via OGT-mediated O-GlcNAcylation of the TET protein family.

# Materials and Methods

## Cell lines

The Flp-In T-REx HEK293 cell line was purchased from Invitrogen (R78007; Invitrogen). *PROSER1* KO and *FLAG-HA-mNeonGreen* knock-in *PROSER1* cells were generated from wild-type Flp-In T-REx HEK293 cells by the Center for Advanced Genome Engineering (CAGE) at St. Jude Children's Research Hospital using CRISPR/Cas9-mediated gene editing. Isogenic tetracycline-inducible FLAG-UTX, FLAG-PROSER1, and FLAG control HEK293 cells were generated by using Flp recombinase-mediated integration. All cells were cultured in Dulbecco's Modified Eagle Medium (11995065; Gibco) with 10% FBS (97068-085; VWR) and 1% Penicillin/Streptomycin (15140122; Gibco).

## Plasmids and molecular cloning

*pcDNA3-FLAG-mTET2-FL* (full length), *pcDNA3-FLAG-mTET2-N500*, *pcDNA3-FLAG-mTET2-N1041*, and *pcDNA3-FLAG-mTET2-CD* were purchased from Addgene (deposited by the Xiong lab). *pcDNA3-FLAG-hTET2-Cys-rich*, *pcDNA3-FLAG-hTET2-DSBH1*, *pcDNA3-FLAG-hTET2-LCI*, *pcDNA3-FLAG-hTET2-DSBH2*, *pcDNA3-FLAG-hTET2-DSBH2-N*, and *pcDNA3-FLAG-hTET2-DSBH2-C* were cloned using the primers shown in Table S2. *hTET2* fragments were PCR-amplified from cDNA of HEK293 cells and the *pcDNA3* backbone was PCR-amplified from *pcDNA3-Flag-hDPY30* obtained from Addgene (deposited by the Ge lab). The *hTET2* fragments and *pCDNA3* backbone were ligated using a Gibson Assembly Master Mix (E2611L; NEB). cDNA was synthesized using the PrimeScript IV first strand cDNA Synthesis Mix from Takara (6215A) after RNA extraction from HEK293 cells. PCR reactions were performed using the Q5 High-Fidelity 2X Master Mix from NEB (M0492L).

## Plasmid transfection

Plasmid transfection was carried out using Lipofectamine 2000 Transfection Reagent (11668019; Invitrogen) according to the manufacturer's instructions. Briefly, cells were plated at 70–90% confluency and transfections were carried out by resuspending the relevant plasmid DNA-lipid complexes in Opti-MEM Reduced Serum Medium (31985070; Gibco), followed by incubation at RT for 10 min. The resuspended DNA-lipid complexes were then added in a dropwise manner to the plated cells followed by gentle mixing on a rotator at RT for 5 min. The cells were then incubated for 2–3 d at 37°C.

## Antibodies

### For Western blotting

Mouse $\alpha$-Actin (JLA20 supernatant; Developmental Studies Hybridoma Bank) at 1:1,000; rabbit $\alpha$-DNMT1 (5032S; Cell Signaling Technology) at 1:2,000; mouse $\alpha$-DNMT3A (sc-373905; Santa Cruz) at 1:500; rabbit $\alpha$-DNMT3B (PA5-85549; Invitrogen) at 1:2,000; mouse $\alpha$-FLAG (F3165; Sigma-Aldrich) at 1:2,000; mouse $\alpha$-FLAG HRP conjugated (A8592; Sigma-Aldrich) at 1:2,000; rabbit $\alpha$-MLL3/KMT2C (#31865 and #31866; Herz Lab [both human aa 581–850]) at 1:5,000; rabbit $\alpha$-MLL4/KMT2D (#31863; Herz Lab [human aa 1–181] and #32757 [human aa 281–506] at 1:5,000; rabbit $\alpha$-OGT HRP conjugated [23177S; Cell Signaling Technology] at 1:2,000; rabbit $\alpha$-OGT [sc-32921; Santa Cruz] at 1:1,000; mouse $\alpha$-O-Linked N-Acetylglucosamine HRP conjugated [ab201995; Abcam]; rabbit $\alpha$-PROSER1 [#34429; Herz Lab] [human aa 302–597], #34641 [human aa 2–233], #34643 [mouse aa 582–811]), serum, at 1:5,000; rabbit $\alpha$-RBBP5 (13171S; Cell Signaling Technology) at 1:2,000; mouse $\alpha$-TET1 (MA5-16312; Invitrogen) at 1:2,000; rabbit $\alpha$-TET2 (A304-247A; Bethyl Laboratories) at 1:2,000; rabbit $\alpha$-TET2 (18950S; Cell Signaling Technology) at 1:2,000; rabbit $\alpha$-UTX/KDM6A (33510S; Cell Signaling Technology) at 1:2,000.

### For dot blot analysis

Mouse $\alpha$-5mC (A-1014-050; Epigentek) at 1:2,000; mouse $\alpha$-5hmC (A-1018-050; Epigentek) at 1:2,000; rabbit $\alpha$-5fC (74178S; Cell Signaling Technology) at 1:2,000; rabbit $\alpha$-5caC (36836S; Cell Signaling Technology) at 1:2,000.

### For ChIP-seq

Rabbit $\alpha$-H3K4me1 (31-1046-00; RevMAb), 10 $\mu$g per ChIP; mouse $\alpha$-H3K4me2 (39679; Active Motif), 10 $\mu$g per ChIP; rabbit $\alpha$-H3K4me3 (31-1039-00; RevMAb), 10 $\mu$g per ChIP; rabbit $\alpha$-H3K27ac (31-1056-00; RevMAb), 10 $\mu$g per ChIP; rabbit $\alpha$-OGT (sc-32921; Santa Cruz), 10 $\mu$g per ChIP; 10 $\mu$g per ChIP; rabbit $\alpha$-OGT (61355; Active Motif), 10 $\mu$g per ChIP; rabbit $\alpha$-PROSER1 (#34429; Herz Lab, human aa 302–597), purified, 10 $\mu$g per ChIP; rabbit $\alpha$-PROSER1 (#34641; Herz Lab, human aa 2–233), purified, 10 $\mu$g per ChIP; rabbit $\alpha$-PROSER (#34643; Herz Lab, mouse aa 582–811), purified, 10 $\mu$g per ChIP; mouse $\alpha$-TET1 (MA5-16312; Invitrogen), 10 $\mu$g per ChIP; rabbit $\alpha$-TET2 (A304-247A; Bethyl Laboratories), 10 $\mu$g per ChIP; rabbit $\alpha$-TET2 (18950S; Cell Signaling Technology), 30 $\mu$l per ChIP; rabbit $\alpha$-UTX/KDM6A (A302-374A; Bethyl Laboratories), 10 $\mu$g per ChIP; rabbit $\alpha$-UTX/KDM6A (33510S; Cell Signaling Technology); 30 $\mu$l per ChIP.

### For hMeDIP-seq

Rabbit $\alpha$-5-hmC (39791; Active Motif), 10 $\mu$g per hMeDIP.

## Immunoprecipitation (IP)

### For large scale IP

Cells at 90–95% confluency from 10 150 mm plates were scraped off with a cell scraper and pelleted by centrifugation. The pellet was washed once with cold DPBS, centrifuged, resuspended in five pellet volumes of Buffer A and incubated on ice for 15 min. After centrifugation, the pellet was resuspended in two pellet volumes of Buffer A. Cytoplasmic lysis was carried out by homogenization with a 15 ml Dounce Tissue Grinder with 15 strokes using a tight pestle (357544; Wheaton). The nuclei were centrifuged for 5 min at 1,500$g$ at 4°C and then resuspended in Buffer C and adjusted to a total volume of 5 ml. Nuclear proteins were extracted by incubation on a nutator for 30 min at 4°C and centrifuged for 30 min at 50,000 rpm at 4°C using an ultracentrifuge (rotor SW 55 Ti; Beckman). The protein concentrations of the nuclear extracts were determined by a protein assay (5000006; Bio-Rad) with a spectrophotometer (6133000010; Eppendorf) using disposable cuvettes (0030079353; Eppendorf). Equal total protein amounts were used for all samples of the same IP experiment. Preclearing was conducted with 200 $\mu$l packed Protein A/G Plus Agarose beads (sc-2003; Santa Cruz) per IP sample (from 0.8 ml bead slurry) by incubation on a nutator for 1 h at 4°C. After centrifugation for 1 min at 850$g$ at 4°C, the precleared supernatant was collected and a small volume was set aside as input. FLAG affinity purifications were carried out with 50 $\mu$l packed (100 $\mu$l bead slurry) FLAG M2 affinity gel (A2220; Sigma-Aldrich) or FLAG magnetic beads (A36797; Pierce) per IP sample on a nutator overnight at 4°C. On the next day the beads were centrifuged for 1 min at 850$g$ at 4°C or separated using a magnetic stand (12321D; Thermo Fisher Scientific). The beads were washed three times with Wash Buffer and one time with Elution Base Buffer. Proteins were eluted by adding 400 $\mu$l Elution Base Buffer supplemented with 200 $\mu$g/ml FLAG peptide (F3290; Sigma-Aldrich) to the FLAG agarose beads or FLAG magnetic beads followed by incubation with mixing on a nutator for 30 min at 4°C. The eluates including the FLAG agarose beads were transferred onto a Micro Bio-Spin column (89868; Pierce) and collected by centrifugation for 2 min at 850$g$ at 4°C. For the FLAG magnetic beads, the elutes were collected by using a magnetic stand. The flow-through (eluates) was saved for further processing.

### For small scale IP

Cells at 90–95% confluency from one 100 or 150 mm plate were scraped off with a cell scraper and pelleted by centrifugation. The pellet was washed once with DPBS. Cytoplasmic lysis was carried out by resuspending the cell pellet with 1 ml Buffer A supplemented with 0.5% NP-40 and incubated on a nutator for 5 min at 4°C. The nuclei were centrifuged for 5 min at 1,500$g$ at 4°C and then resuspended in Buffer C. If using total cell lysates, the cell pellets were directly lysed in Buffer C supplemented with 0.5% NP-40. Protein extraction was performed by incubation on a nutator for 30 min at 4°C and centrifugation for 30 min at 20,817$g$ at 4°C using a table top centrifuge (5430R; Eppendorf). Equal protein amounts were used for all samples of the same IP experiment. For FLAG IPs, 50 $\mu$l FLAG magnetic bead slurry (A36797; Pierce) was used. For TET2 IPs, 50 $\mu$l Dynabeads Protein G slurry (10004D; Thermo Fisher Scientific) and 5 $\mu$g TET2 antibody (A304-247A; Bethyl Laboratories) were used. Incubation was performed on a nutator at 4°C for at least 2 h. After the incubation, the

beads were collected with a magnetic stand and subjected to washing. The washed beads were then boiled in 1× SDS Laemmli Buffer for 10 min at 95°C.

### Buffer A

10 mM Hepes, pH 7.9; 10 mM KCl; 1.5 mM MgCl$_2$; 0.5 mM DTT; and protease inhibitors (P8340; Sigma-Aldrich) (1:200).

### Buffer C

20 mM Hepes, pH 7.9; 420 mM NaCl; 1.5 mM MgCl$_2$; 0.2 mM EDTA; 10% glycerol; 0.5 mM DTT; and protease inhibitors (P8340; Sigma-Aldrich) (1:200).

### Wash Buffer

20 mM Hepes, pH 7.9; 300 mM NaCl; 1.5 mM MgCl$_2$; 10% glycerol; 0.1% Triton X-100; 0.5 mM DTT; and protease inhibitors (P8340; Sigma-Aldrich) (1:500).

### Elution Base Buffer

20 mM Hepes, pH 7.9; 100 mM NaCl; 1.5 mM MgCl$_2$; 10% glycerol; 0.05% Triton X-100; 0.5 mM DTT; and protease inhibitors (P8340; Sigma-Aldrich) (1:500).

### 4× SDS Laemmli Buffer

250 mM Tris, pH 6.8; 50% glycerol; 8% SDS; and 0.1% bromophenol blue. Before using, add 100 $\mu$l of $\beta$-mercaptoethanol to 900 $\mu$l of 4× buffer.

## Glycerol gradient fractionation

The eluates of two large scale FLAG affinity purifications of FLAG-UTX from HEK293 cells were combined and brought up to 1 ml by addition of Elution Base Buffer. The eluates were then layered over 10 ml of a 20–50% glycerol gradient and centrifuged for 48 h at 35,000 rpm at 4°C using an ultracentrifuge (rotor SW 41 Ti; Beckman). 33 fractions of ~325 $\mu$l each were collected and analyzed by WB for UTX, RBBP5, PROSER1, and TET2.

## RNA-seq

RNA was isolated from 3 × 10$^6$ cells with the RNeasy Mini Kit (74106; QIAGEN) following the manufacturer's instructions with the following alterations. Cells were homogenized in 600 $\mu$l RLT buffer (with 2-Mercaptoethanol) and passed through a QIAshredder column (79656; QIAGEN) by centrifugation in a table top centrifuge for 2 min at full speed at RT. The optional step after the second wash with RPE buffer was applied to dry the membrane. For DNA digestion, RNA was eluted with 85 $\mu$l H$_2$O and mixed with 10 $\mu$l 10× DNAase buffer and 5 $\mu$l DNAse I (M0303S; NEB), then incubated at RT for 20 min. After incubation, RNA was purified by following the "RNA Cleanup" protocol in the RNeasy Mini Handbook. RNA was finally eluted in 50 $\mu$l H$_2$O and concentration determined with a NanoDrop 8000 Spectrophotometer. The following RNA quantification, quality check, strand-specific library preparation, and sequencing were performed by the Hartwell Center at St. Jude Children's Research Hospital as previously described (Mondal et al, 2020).

## RNA-seq data processing

Sequencing reads were quality-filtered using TrimGalore (https://www.bioinformatics.babraham.ac.uk/projects/trim_galore/). Filtered reads were aligned to GRCh38 using STAR (Dobin et al, 2013). RSEM was used to quantify read counts per gene (Li & Dewey, 2011). Differential expression analysis was assessed using limma-voom (Law et al, 2014; Ritchie et al, 2015). Only level 1 and 2 protein-coding genes, with at least 10 reads per sample in the minimum group size, were retained in the analysis.

## hMeDIP-seq

hMeDIP was performed by using the hMeDIP Kit (55010; Active Motif) according to the provided instruction manual. Briefly, 20 $\mu$g genomic DNA in 300 $\mu$l 10 nM Tris–HCl, pH 8.5 buffer was sonicated to 200–600 bp using a Bioruptor Plus sonication device (B01020001; Diagenode). 100 ng of fragmented DNA was set aside as input DNA. A 100 $\mu$l mixture was set up containing 1 $\mu$g sonicated DNA, 1 $\mu$l protease inhibitor cocktail, 10 $\mu$l Buffer C, and 4 $\mu$l $\alpha$-5-hydroxy-methylcytidine antibody. The mixture was incubated overnight with end-to-end rotation at 4°C. On the next day, 25 $\mu$l Protein G magnetic beads were added to each tube followed by incubation for 2 h with end-to-end rotation at 4°C. The beads were collected using the provided magnetic stand, washed three times with ice cold Buffer C and then two times with ice cold Buffer D. The washed beads were resuspended in 50 $\mu$l Elution Buffer AM2 and incubated for 15 min at RT with end-to-end rotation. 50 $\mu$l Neutralization Buffer was added the sample mixed with a pipette. The beads were collected with a magnetic stand and the supernatant transferred to a fresh tube. DNA from all inputs and supernatants was purified using the QIAquick PCR Purification Kit (28106; QIAGEN). DNA quantification, quality check, library preparation and sequencing were performed by the Hartwell Center at St. Jude Children's Research Hospital as previously described (Mondal et al, 2020).

## Dot blot

Genomic DNA was extracted from cells using the QIAamp DSP DNA Mini Kit (61304; QIAGEN). After DNA was denatured with NaOH at RT for 10 min, 1,000, 500, 250, and 125 ng of DNA was blotted onto a nitrocellulose membrane (1228243; GVS). DNA was cross-linked twice using the Stratalinker UV Crosslinker (400071; Stratagene) under "AUTO CROSS LINK" mode. The membrane was blocked for 1 h at RT with gentle rocking in TBST buffer supplemented with 5% nonfat dry milk. Antibody incubation (1:2,000 dilution) was performed overnight at 4°C with gentle rocking in TBST buffer supplemented with 1% nonfat dry milk. On the next day, the membrane was washed three times for 10 min with 10 ml TBST buffer with gentle rocking and then incubated with a horse radish peroxidase coupled secondary IgG-specific antibody in TBST buffer supplemented with 1% nonfat dry milk for 1 h at RT with gentle rocking. After three additional washes for 10 min with 10 ml TBST buffer with gentle rocking, the membrane was developed using Immobilon Crescendo Western HRP Substrate (WBLUR0500; Millipore) and imaged on an Odyssey Fc imaging system (Model: 2800; LI-COR). After imaging, the membrane was stained with Methylene Blue to show the presence of DNA.

## ChIP-seq

ChIPs were performed according to a modified version of Lee et al (2006) as reported previously (Lee et al, 2006; Mondal et al, 2020). ChIPs for non-histone proteins were carried out with 5 × 10$^7$ HEK293 cells and for histone marks with 2.5 × 10$^7$ HEK293 cells. Dual cross-linking was performed at RT for 30 min with 2 mM disuccinimidyl glutarate (DSG) in DPBS followed by addition of paraformaldehyde to a final concentration of 1% for another 15 min. Cross-linking was quenched with 150 mM glycine for 5 min at RT. After sequential lysis of the cros-slinked cells in Lysis Buffer 1 and Lysis Buffer 2, the nuclear pellet was sonicated in Lysis Buffer 3 with a probe sonicator (Model 705 Sonic Dismembrator; Thermo Fisher Scientific) at output level 55 (27–33 W) for 16 cycles with each cycle constituting a 30 s sonication burst followed by a 60-s pause. For each ChIP, 100 $\mu$l Protein G Dynabead (10003D; Invitrogen) slurry and 10 $\mu$g of antibody was used. Chromatin incubation with antibodies was carried out at 4°C with end-to-end rotation for 4–6 h. The remainder of the protocol was carried out as described previously (Mondal et al, 2020). ChIP-seq library preparation and sequencing was carried out by the Hartwell Center at St. Jude Children's Research Hospital as described previously (Mondal et al, 2020).

## ChIP-seq and hMeDIP-seq data processing

Raw reads in fastq format were processed with the Trim-Galore tool (v0.4.4, https://www.bioinformatics.babraham.ac.uk/projects/trim_galore/) (Krueger et al, 2012), potential adapters were removed and the 3' end of reads quality trimmed with cutadapt (DOI: 10.14806/ej.17.1.200), followed by FastQC analysis with quality score cutoff of Q20. FastQC: a quality control tool for high throughput sequence data. Available online at: http://www.bioinformatics.babraham.ac.uk/projects/fastqc. Next, reads were mapped to the human reference genome (GRCh38.p12) with bwa aln, followed by bwa samse (Li & Durbin, 2009) (v0.7.12-r1039) with -K flag set to 10,000,000 followed and the output converted to binary alignment map (BAM) format with samtools (Li et al, 2009) (v1.2). Afterwards, duplicated reads were marked with the bamsormadup tool from biobambam2 (v2.0.87, DOI: 10.1186/1751-0473-9-13) and Cross-Correlation analysis was conducted with SPP (Kharchenko et al, 2008) (v1.11). Uniquely mapped reads were then extracted with samtools, extended with bedtools (Quinlan & Hall, 2010) (v2.24.0) using the fragment size value previously estimated by Cross-Correlation analysis, and then converted to bigwig track files by University of California, Santa Cruz (UCSC) tools (Kuhn et al, 2013) (v4). Subsequently, MACS2 (Zhang et al, 2008) was used to call peaks in narrow mode, with –nomodel -q 0.05 flags (high confidence peaks). Separately, narrow peaks were also called with more relaxed criteria, setting the -q flag to 0.5, which are here referred to as FDR50 peaks. Next, for experiments with more than one replicate, reproducible peaks were identified as those with overlapping FDR50 peaks present in all replicates at a given genomic region. Otherwise, for ChIP-seq targets without replicates, only the high-quality peaks were considered. Then, using in-house scripts, raw read counts per peak were transformed to reads per kilobase per million mapped reads (RPKM). Afterwards, the fold-change between targets (*PROSER1* KO samples) and their paired controls (WT samples) was computed separately for each antibody subtype. Regions with fold-change higher

than two were considered as differentially binding. Finally, if more than one antibody was used for a particular protein (e.g., PROSER1), the union of up-regulated and down-regulated peaks was used as the final set of differentially bound regions. For hMeDIP-seq data the ChIP-seqSpikeInFree tool (Jin et al, 2020) was used to determine the presence or absence of genome-wide loss of 5hmC signal. In parallel, for visualization purposes, the bigwig tracks were re-normalized within the same type of target (e.g., all WT and *PROSER1* KO samples with antibodies targeting PROSER1), dividing the enrichment values of the bigwig files by the median value of the signal collected only from the overlap of high confidence peaks between replicates. Subsequently the same phenotypes (e.g., PROSER1 WT) were merged computing the average signal between samples. These tracks were then used with deeptools (Ramirez et al, 2014) to generate the heatmaps of the enrichment of various proteins and histone modifications.

## WGBS

WGBS was carried out by the Hartwell Center at St. Jude Children's Research Hospital. Briefly, genomic DNA was extracted using the QIAamp DSP DNA Mini Kit (61304; QIAGEN) and then was bisulfite converted using the EZ-96 DNA Methylation-Gold MagPrep Kit (D5042; Zymo Research). Libraries were prepared from converted DNA using the Accel-NGS Methyl-Seq DNA Library Kit (30096; Swift Biosciences). Libraries were analyzed for insert size distribution with a 2100 Bio-Analyzer High Sensitivity Kit (Agilent Technologies), 4200 TapeStation D1000 ScreenTape assay, or Caliper LabChip GX DNA High Sensitivity Reagent Kit (PerkinElmer). Libraries were quantified using the Quant-iT PicoGreen ds DNA assay (Life Technologies) or low pass sequencing with a MiSeq nano kit (Illumina). Paired-end 150 cycle sequencing was performed on a NovaSeq 6000 (Illumina).

## WGBS data processing

Raw bisulfite converted reads were processed with the Trim-Galore tool (v0.6.5, https://www.bioinformatics.babraham.ac.uk/projects/trim_galore/) (Krueger et al, 2012). The following settings were applied: -q 30 –fastqc –phred33 –illumina –stringency 1 -e 0.1 –paired –length 15 –clip_R1 10 –clip_R2 10 –three_prime_clip_R1 10 –three_prime_clip_R2 10. Next, BSMAP (Xi & Li, 2009) (v2.9.0) was used to map bisulfite reads to the human reference genome (GRCh38.p12) with the following settings: -m 17 -x 600 -z 33 -f 5 -g 3 -r 0 -u -R; and the output was converted to BAM format with samtools (Li et al, 2009) (v1.4). Duplicated reads were marked with the Picard toolkit (v2.0.1, http://broadinstitute.github.io/picard/). Cytosine methylation ratios in CG context were extracted from BAM files using the methratio.py script from BSMAP (https://github.molgen.mpg.de/molgen/bsmap/blob/master/methratio.py). methylKit (Akalin et al, 2012) (v1.12.0) was used to identify differentially methylated cytosines, defined as those with a minimal methylation difference of 25% and false discovery rate (FDR) threshold set to 0.05. Only cytosines covered by at least 10 reads in all samples were included in the analysis. Moreover, the same criteria were also independently applied for the identification of several types of differentially methylated regions in *PROSER1* KO versus WT cells: (1) CpG islands (CGIs), downloaded from UCSC (http://hgdownload.cse.ucsc.edu/goldenpath/hg38/database/cpgIslandExt.txt.gz); (2) CGI shores (2 kbp regions flanking CGIs); (3) promoter regions (TSS ± 2 kbp); as well as genomic bins of (4) 100,

(5) 500 and (6) 1,000 bp long windows, sliding by half their length; (7) TET1 down-regulated peaks; (8) TET2 down-regulated peaks; (9) TET1 up-regulated peaks; (10) TET2 up-regulated peaks; (11) peaks with decreased occupancy of both UTX and H3K4me1; (12) peaks with decreased occupancy of PROSER1. Independently, CpGs covered by at least four reads per replicate were extracted for the identification of CGI encroachment, as described in the subsequent paragraph.

## Identification of CGI encroachment events

Identification of 5' end and 3' end encroachment events, inspired by research of Skvortsova et al (2019), was conducted with in-house scripts following the procedure described in detail in the following paragraph. First, the human genome was separated into 40-bp long bins, containing the average methylation signal of all overlapping CpGs. This binarization was based on the methylation ratios of CpGs covered by at least four sequencing reads in WT and *PROSER1* KO samples. Next, the methylation difference was computed by subtraction of the per-bin methylation level in *PROSER1* KO from the corresponding bin in WT samples. Bins not having a methylation level in both conditions were discarded from further analysis. Then, using bedtools, those CGIs from among all CGIs were extracted that were overlapping with at least four bins showing at least a 25% increase in *PROSER1* KO compared with WT samples, additionally requiring at least one of them to show at least a 50% increase compared with the WT sample. Next, those prefiltered CGIs were overlapped with CpG-level, not bin-level, methylation ratios. Regardless of the original CGI length, each CGI was then separated into 40 segments; and for each of those segments the average methylation level was computed based on the methylation level of the CpGs falling into that segment. If no CpGs were falling into a particular segment, the average methylation level was replaced with a "Not-a-Number" (NaN) value. These steps are conceptualized in Fig S8 which displays a simplified example of CGI separation into five segments. Next, segmented methylation values of each CGI from *PROSER1* KO samples were subtracted from their counterparts in WT samples, the result of which is referred to here as differential methylation matrix. In case such an operation was not possible because of the presence of a NaN value in either condition, a NaN value was placed in the resulting differential methylation matrix. All CGIs which after these steps consisted of more than 30 NaN values, were excluded from further analysis. Finally, CGIs retained in the differential methylation matrix were ranked by their average methylation value derived from the first 10, middle 20 and last 10 segments, which represents 5' end hypermethylation (5' end encroachment), central hypermethylation (i.e., either bidirectional encroachment or full CGI hypermethylation) and 3' hypermethylation (3' end encroachment), respectively. The top 250 CGIs were selected from each of those rankings as representing various scenarios and degrees of encroachment, which were subsequently subjected to further study (Table S1).

## Annotation of genomic regions

Genomic regions were assigned to their genomic context with an in-house script based on pybedtools (Dale et al, 2011) (v0.8.1), such that each region could only be assigned to one feature. For this purpose, genomic regions were successively overlapped with predefined genomic contexts in the following order: (1) Promoter.Up: region up to 2 kbp upstream from TSS; (2) Promoter.Down: region up to 2 kbp downstream

from TSS; (3) Exons; (4) Introns; (5) TES–transcription end sites; (6) 5′ Distal: region up to 50 kbp upstream from TSS, excluding promoter region; (7) 3′ Distal: region up to 50 kbp downstream from TSS, excluding promoter region; (8) Intergenic. In addition, to establish the background genome-wide distribution of features, the reference human genome was separated into 1 kbp long bins with the makewindows tool from bedtools. The reference annotation for TSS, and all subsequent genomic contexts, was based on the Gencode v31 (Frankish et al, 2019) reference annotation including all isoforms. To identify the specific genes associated and potentially influenced by differentially bound genomic regions, peaks were associated with all genes whose promoter regions (combination of both Promoter.Up and Promoter.Down) they were overlapping with or could be assigned to. Here one genomic region could have been assigned to multiple genes. The same strategy was used to assign also differentially methylated regions (e.g., CGIs).

### CRISPR/Cas9 gene editing

Flp-In T-REx HEK293 *PROSER1* KO and Flp-In T-REx HEK293 *FLAG-HA-mNeonGreen* tagged *PROSER1* (*FHNG-PROSER1*) cells were generated using CRISPR/Cas9 technology. Briefly, ≈400,000 Flp-In T-REx HEK293 cells were transiently co-transfected with precomplexed RNPs consisting of 100 pmol of chemically modified sgRNA (Synthego) and 35 pmol of Cas9 protein (St. Jude Protein Production Core) (Table S3). In addition, 200 ng of pMaxGFP (Lonza) or 1 μg of donor plasmid were also included in the transfection mixes for either the KO or tagged lines, respectively. The transfection was performed via nucleofection (4D-Nucleofector X-unit; Lonza) using solution P3 and program CM-130 in a small (20 μl) cuvette according to the manufacturer's recommended protocol. 5 d post nucleofection, cells were single cell sorted for GFP+ (transfected or tagged) cells by FACs in 96-well plates and clonally selected. KO clones were screened and verified for the desired out-of-frame modification via targeted deep sequencing using gene specific primers with partial Illumina adapter overhangs as previously described (Table S3) (Sentmanat et al, 2018). In brief, clonal cell pellets were harvested, lysed and used to generate gene specific amplicons with partial Illumina adapters in PCR#1. Amplicons were indexed in PCR#2 and pooled with other targeted amplicons for other loci to create sequence diversity. In addition, 10% PhiX Sequencing Control V3 (Illumina) was added to the pooled amplicon library before running the sample on an Miseq Sequencer System (Illumina) to generate paired 2 × 250 bp reads. Samples were demultiplexed using the index sequences, fastq files were generated, and NGS analysis was performed using CRIS.py (Connelly & Pruett-Miller, 2019). Tagged clones were screened for the targeted integration event using primers CAGE197.5gen.F and CAGE197.5junc.R to the 5′ junction and CAGE197.3junc.F and CAGE197.3gen.R to the 3′ junction (Table S3). Junctions were sequence confirmed. Final clones were authenticated using the PowerPlex Fusion System (Promega) performed at the Hartwell Center (St. Jude) and tested negative for mycoplasma by the MycoAlert Plus Mycoplasma Detection Kit (Lonza).

## Data Availability

The accession number for the RNA-seq, ChIP-seq, hMeDIP-seq, and WGBS-seq datasets reported in this article is Gene Expression Omnibus: GSE172145.

## Supplementary Information

## Acknowledgements

We thank the Yue lab for providing mouse TET2 deletion constructs. We are grateful for the following research resources at St. Jude Children's Research Hospital: the Hartwell Center for RNA-seq, ChIP-seq, hMeDIP-seq, and WGBS library preparation and sequencing, and the center for Proteomics and Metabolomics for conducting mass spectrometry analysis. The mouse α-Actin monoclonal hybridoma antibody (JLA20) developed by the University of Iowa was obtained from the Developmental Studies Hybridoma Bank, created by the Eunice Kennedy Shriver National Institute of Child Health and Human Development (NICHD) of the National Institutes of Health (NIH) and maintained at The University of Iowa, Department of Biology, Iowa City, IA 52242. We thank all Herz lab members for insightful comments and critical reading of the manuscript. This work was supported by a transition to independence grant from the National Institutes of Health/National Cancer Institute (R00CA181506 to H-M Herz, P30CA021765 to SM Pruett-Miller); and the American Lebanese Syrian Associated Charities (ALSAC). K Helin was supported by the Independent Research Fund Denmark (8020-00044) and through a center grant from the Novo Nordisk Foundation to the Novo Nordisk Foundation (NNF) Center for Stem Cell Biology (NNF17CC0027852). The content is solely the responsibility of the authors and does not necessarily represent the official views of the National Institutes of Health.

### Author Contributions

X Wang: conceptualization, data curation, formal analysis, validation, investigation, visualization, methodology, project administration, and writing—original draft, review, and editing.

W Rosikiewicz: data curation, software, formal analysis, investigation, visualization, methodology, and writing—review and editing.

Y Sedkov: data curation, formal analysis, validation, investigation, visualization, and methodology.

T Martinez: data curation, formal analysis, validation, investigation, and methodology.

BS Hansen: resources, data curation, formal analysis, validation, investigation, visualization, and methodology.

P Schreiner: data curation, software, formal analysis, validation, visualization, and methodology.

J Christensen: resources.

B Xu: data curation, software, formal analysis, visualization, and methodology.

SM Pruett-Miller: resources, data curation, formal analysis, supervision, funding acquisition, validation, investigation, visualization, methodology, and project administration.

K Helin: resources and funding acquisition.

H-M: Herz: conceptualization, resources, data curation, formal analysis, supervision, funding acquisition, validation, investigation, visualization, methodology, project administration, and writing—original draft, review, and editing.

### Conflict of Interest Statement

K Helin is a consultant for Inthera Bioscience AG and a scientific advisor for Hannibal Health Innovation. All other authors declare no competing interests.

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
