## [Reviewer comments · Life Science Alliance]

Life Science Alliance

PROSER1 mediates TET2 O-GlcNAcylation to regulate DNA demethylation on UTX-dependent enhancers and CpG islands

Xiaokang Wang, Wojciech Rosikiewicz, Yurii Sedkov, Tanner Martinez, Baranda Hansen, Patrick Schreiner, Jesper Christensen, Beisi Xu, Shondra Pruett-Miller, Kristian Helin, and Hans-Martin Herz

DOI: <https://doi.org/10.26508/lsa.202101228>

Corresponding author(s): Hans-Martin Herz, St. Jude Children's Research Hospital

Review Timeline:

Submission Date:	2021-09-07
Editorial Decision:	2021-09-08
Revision Received:	2021-09-28
Editorial Decision:	2021-09-30
Revision Received:	2021-10-01
Accepted:	2021-10-04

Transaction Report:

Please note that the manuscript was reviewed at *Review Commons* and these reports were taken into account in the decision-making process at *Life Science Alliance*.

September 8, 2021

Re: Life Science Alliance manuscript #LSA-2021-01228

Hans-Martin Herz
St. Jude Children's Research Hospital
262 Danny Thomas Place
Memphis, TN 38105-3678

Dear Dr. Herz,

Thank you for submitting your manuscript entitled "PROSER1 mediates TET2 O-GlcNAcylation to control DNA demethylation on UTX-dependent enhancers and CpG islands" to Life Science Alliance. We invite you to re-submit the manuscript, revised according to your Revision Plan.

Thank you for this interesting contribution to Life Science Alliance. We are looking forward to receiving your revised manuscript.

Sincerely,

B. MANUSCRIPT ORGANIZATION AND FORMATTING:

Response to Reviewers

Reviewer #1 (Evidence, reproducibility and clarity (Required)):

This paper shows that TET2, OGT and PROSER1 all interact with UTX, a component of the enhancer-associated MLL3/4 complexes, which are H3K4 monomethyltransferases. They show that PROSER1 mediates the interaction between OGT and TET2 promoting TET2 OGN and protein stability. The four proteins co-localize on many genomic elements genome wide. They also demonstrate that loss of PROSER1 reduces these complexes at enhancers and CpG islands with concomitant increased DNA methylation. Finally, the authors also suggest that PROSER1 might be a more general regulator of OGT activity by controlling OGN in other chromatin signaling pathways. Overall, the data support the author's conclusions. The paper is well written and the data are clear. One figure, Figure 2G is not the most convincing given the unidentified closely migrating bands, but overall, combined with the rest of the data, this is not a major concern.

Reviewer #1 (Significance (Required)):

Overall this paper is a significant advance toward our understanding of the molecular regulation of transcription and the identification of PROSER1's roles in regulating chromatin modifications is highly significant. I would give this paper a high priority.

****Referee Cross-commenting****

Reviewer #2s comments are valid, but the request to measure site occupancy on TET2 by O-GlcNAc is very difficult to actually perform and while I think it would be great, it is not essential to the major conclusions.

Reviewer #2 (Evidence, reproducibility and clarity (Required)):

Genomic methylation patterns have been heavily studied but the mechanism for their establishment is still partially understood. The work presented in this manuscript reports the functional identification of a new player, PROSER1, for the regulation of DNA methylation. PROSER1 is shown to form a functional group with UTX, OGT and TET2. PROSER1 promotes the O-GlcNAcylation of the TET2 protein by OGT which in turn increases the TET2 protein stability and stimulates the demethylation activity of the TET2 enzyme. Importantly, the authors show the four interacting components have overlapping genomic distributions and the loss of PROSER1 compromises the genome binding of the other three components, leading to the spread of DNA methylation into otherwise unmethylated CpG island regions. The findings to this point are already novel and significant. However, I would suggest a few extra experiments to gain more support for the conclusion stated in the last sentence of the abstract. As an alternative, the conclusion can be toned down and be more specific about the findings presented.

We thank the reviewers for providing us with a timely review of our manuscript. We are grateful that both reviewers are of the conviction that our manuscript presents a significant advance to the field by providing vital insight into the role of the previously undescribed proline and serine rich protein PROSER1 in regulating TET2 protein O-GlcNAcylation and stability to control transcription via DNA demethylation on UTX-dependent enhancers and CpG islands. Reviewer 1 gives our manuscript a high priority and does not ask for any extra work to be done. While reviewer 2 thinks that our findings are already novel and significant in themselves, he/she raises several plausible concerns which would require additional experimental evaluation and bioinformatics analysis but also states that alternatively some of the conclusions in the manuscript could be toned down without requiring any additional experiments. As Reviewer 1 did not request any additional experiments and bioinformatics analysis, here we only proceed with a point-by-point response to the questions raised by Reviewer 2.

1. For the characterization of the biological significance of the interaction between TET2 and PROSER1, experiments would be required to specifically disrupt the protein-protein interaction using the endogenous proteins of HEK293 cells. Since the 66 amino acid region of TET2 mapped to mediate the interaction with PROSER1 is located in the catalytic domain of TET2, care should be taken so that a specific mutation introduced to disrupt the interaction should not interfere with the enzymatic activity of TET2.

Answer: We thank the reviewer for raising this question and agree that it would be interesting to identify mutations within the 66 amino acid region of TET2 that disrupt its interaction with PROSER1 and OGT. Hu et al., 2013 (Hu et al 2013), who tested the catalytic activities of various N-terminal and C-terminal truncation constructs of TET2 expressed in HEK293T cells, came to the conclusion that truncation of the N-terminus before residue 1129 or of the C-terminus after residue 1936 had no significant effect on TET2 activity (Fig S1 in Hu et al., 2013 (Hu et al 2013)). The 66 amino acid region of TET2 which interacts with PROSER1 starts at residue 1937 and ends at residue 2002. Thus, it is unlikely that mutations within the 66 amino acid region of TET2 will directly affect TET2 enzymatic activity. According to the St. Jude Pecan database (<https://pecan.stjude.cloud/>), several mutations in both pediatric and adult cancers including recurrent mutations such as P1962L have been reported within the 66 amino acid region of TET2 making it likely that this region is involved in important physiological and cancer-relevant processes (Fig R1). To test how these mutations might affect the interaction between TET2 and PROSER1 at the endogenous protein level as requested by the reviewer, CRISPR/Cas9 gene editing of the TET2 locus is required to specifically introduce several of these point mutations. This would necessitate the establishment of several TET2 mutant cell lines by CRISPR/Cas9 gene editing and by our estimation demand a time horizon and effort that would by far exceed the potential benefit gained from this experiment, specifically as the outcome is hard to predict. However, we very much agree that this is an important point that should be addressed in a future follow up study.

Figure R1. Mutations in the 66 amino acids region of TET2 in both pediatric and adult cancers according to the St. Jude Pecan database.

2. With regard to TET2 O-GlcNAcylation, to what extent at each site in the TET2 protein is modified? What are the impacts of this modification on the TET2 function in vivo? Is the genome-wide distribution of TET2 (ChIP-seq) altered in OGT-knockout cells or most preferably in cells with a catalytically inactive mutation OGT?

Answer: We thank the reviewer for raising this point. Bauer et al., 2015 (Bauer et al 2015), have systematically investigated O-GlcNAcylation sites in TET family proteins. And several websites also provide predictions or annotations of O-GlcNAcylation sites in TET2, including the O-GlcNAc Database (<https://www.oglcnac.mcw.edu/>) and PhosphoSite (<https://www.phosphosite.org/homeAction>). These sources have identified tens of O-GlcNAcylation sites in TET2. The study of these identified O-GlcNAcylation sites is additionally complicated by the co-occurrence of phosphorylation (Bauer et al 2015) on the same residues. We are in agreement with reviewer 1 that measuring the degree of TET2 O-GlcNAcylation at specific sites is very difficult and also thank reviewer 2 for excluding this requested experiment in the revision. OGT as the only O-GlcNAc transferase in mammals is essentially required for mammalian cell and embryo growth (Konzman et al 2020, Yang & Qian

2017). Therefore, we are unable to assess the genome-wide distribution of TET2 via ChIP-seq in OGT KO HEK293 cells. While it might be possible to investigate genome-wide TET2 occupancy in HEK293 cells with a catalytically inactivating mutation of OGT, this cell line currently does not exist and it would require extensive efforts and time investment to first create this line in order to carry out this experiment.

3. Hypermethylation encroachment at CpG islands in PROSER1-deficient cells could also be explained by excessive activity of DNA methyltransferases. Authors would need to exclude or confirm the potential involvement of DNMT1, DNMT3A and DNMT3B in PROSER knockout cells. For one thing at least, are the protein levels of DNMTs altered in PROSER knockout cells?

Answer: We very much appreciate the reviewer's suggestion. Our current model favors a loss of the TET proteins at CpG islands to explain the observed hypermethylation encroachment phenotype. This interpretation is supported by the data presented in Figures 2 and 3 of our manuscript which shows that PROSER1 prominently interacts with all three members of the TET protein family and is required to stabilize TET1 and TET2 protein levels (Figs 2A and 3D in our revised manuscript) while no association with any member of the DNA methyltransferase (DNMT) family could be detected by mass spectrometry (data not shown). Our interpretation is further corroborated by the strong reduction of TET1 and TET2 and of 5hmC at these CpG islands using 5hmC-IP followed by high throughput sequencing observed in PROSER1 KO cells (Figs S4C and S6B in our revised manuscript). However, we cannot completely rule out a PROSER1-mediated involvement of DNMTs at these sites or on other hypermethylated DMRs or UTX/H3K4me1 downregulated regions. Even if we performed ChIP-seq experiments for various DNMT family members and observed an increase in DNMT enrichment at these sites in PROSER1 KO cells, it would not necessarily argue in favor of a PROSER1-mediated recruitment mechanism of DNMTs. In this case the potential increase in DNMT enrichment could be interpreted as being directly mediated by PROSER1 or alternatively, which is our preference based on the provided evidence in our manuscript, as the indirect result of TET and DNMT family members operating in a dynamic competitive relationship at overlapping target regions with the assumption that if enrichment of a TET family member was reduced at a given locus a DNMT family member might now have easier access. In summary, ChIP-seq applications for DNMT family members will not be able to resolve the question whether PROSER1 is able to mediate DNMT recruitment to CpG island, hypermethylated DMRs or UTX/H3K4me1 downregulated regions. However, as suggested by the reviewer, we have now assessed the mRNA and protein levels of DNMT1, DNMT3A and DNMT3B and did not observe any significant difference between WT and PROSER1 KO cells (Fig S7A in our revised manuscript and data not shown). This suggests that neither on the mRNA nor the protein level DNMT family members are regulated by PROSER1. To raise awareness for the readership we have now expanded the discussion section to specifically discuss the issue of interdependency between the DNA demethylation and methylation machineries.

****Minor comments:****

1. The authors described 'O-GlcNAcylated TET2 could no longer be detected in PROSER1 KO' and 'both TET1 and TET2 protein levels were decreased', but as a consequence, approximately equal numbers of hypermethylated and hypomethylated cytosines were observed. The logical discrepancy would have to be explained. Concerning 'At the 4,421 UTX/H3K4 me1-dependent regions, DNA methylation was increased', the authors didn't mention in which region where DNA methylation was reduced. And do these regions show increased enrichment of Tet2? It is not shown whether 5hmC levels were reduced globally or just at the specific regions (addressing this issue would require ACE-seq or TAB-seq analysis).

It is not fully clear whether Tet2 HEK293 KO cells show a more widely and stronger increase of DNA methylation. Maybe the reduced recruitment of Tet2 doesn't account for the increase of DNA methylation in some regions.

Answer: We appreciate the reviewer's keen eye in noticing that approximately equal numbers of hypermethylated and hypomethylated cytosines were detected in PROSER1 KO cells. Interestingly, this phenomenon has also been observed in TET mutant cells. According to Hon et al., 2014 (Hon et al 2014), in *Tet1*^{-/-} mESCs the vast majority of DMRs (96.1%) are hypomethylation events, while in *Tet2*^{-/-} mESCs DMRs are split more evenly among hypermethylation (31.1%) and hypomethylation (68.9%) events. Lu et al., 2014 (Lu et al 2014) found that Tet triple KO (Tet1 KO, Tet2 KO and Tet3 KO) mESCs showed higher numbers of hypomethylated tiles (651,637) than hypermethylated tiles (524,533). Even in plants with an impaired DNA demethylation machinery such as in *Arabidopsis* mutants, both hyper- and hypomethylated DMRs can be observed (Wang et al 2019, Wang et al 2016). Recently, Lopez-Moyado et al., 2019 (Lopez-Moyado et al 2019) showed that hypomethylation in TET KO cells is largely restricted to heterochromatin while hypermethylation in TET KO cells is mainly confined to euchromatin including active enhancers and gene bodies (including promoters) of actively transcribed genes. This study also reported that in Tet1 KO mESCs the DNA methyltransferase DNMT3A relocalizes from the heterochromatic to the euchromatic compartment providing a potential mechanism for the observed heterochromatin hypomethylation phenotype in TET KO cells. In summary, this points to a functional interaction between the TET protein and DNMT protein families in addition to the dynamic competitive relationship that also exists between these families (discussed in our response to point 3 raised by reviewer 2).

Furthermore, we have now performed additional bioinformatic analyses to address the reviewer's question regarding the hypomethylated loci that can be found in PROSER1 KO cells. The key points from these analyses are as follows:

- a) Upregulated TET1/2 peaks tend to be hypomethylated, while TET1/2 downregulated peaks tend to be hypermethylated in PROSER1 KO versus WT cells (Fig S7B). This indicates that DNA hypomethylation at some regions may result from increased TET enrichment and as mentioned above that the increase in TET enrichment might be the result of relocalized DNMT proteins in PROSER1 KO cells.
- b) Both UTX/H3K4me1 downregulated regions and regions with PROSER1 peaks are strongly and dominantly associated with DNA hypermethylation in PROSER1 KO compared to WT cells (Fig S7B).
- c) We now also ran an additional analysis aimed at determining whether 5hmC is lost globally or only at specific sites in PROSER1 KO cells. For this purpose we utilized the 5hmC measurements from our hMeDIP-seq data that was already included in our manuscript (e.g. Figs 5B and C and S6B). This analysis shows that PROSER1 KO cells exhibit a strong global decrease in 5hmC compared to WT cells (Fig S5D).

TET2 KO HEK293 cells currently do not exist (to our knowledge). Thus, we are not able to directly address the question whether TET2 KO HEK293 cells display a more widespread and stronger increase in DNA methylation than PROSER1 KO cells at this point. Based on what is known about the ratio of hypermethylated to hypomethylated regions in other TET2 KO cell types (see above), it is not to be expected that TET2 KO HEK293 cells will necessarily show a more widespread and stronger increase in DNA methylation than PROSER1 KO cells. Furthermore, our bioinformatics analysis also confirms that the majority of regions that display reduced TET1 or TET2 occupancy in PROSER1 KO cells is associated with DNA hypermethylation (see above). However, we also identified regions with decreased TET1 or TET2 enrichment in PROSER1 KO cells that display decreased DNA methylation. This either points to a compensatory role of TET3 at these regions or as mentioned above to an additional role of DNMT proteins in this context which might have its cause in DNMT relocalization. We have now further elaborated on these possibilities in the discussion section.

2. Since the expression levels of Tet1 and Tet2 are rather low in HEK293 cells. What's the function of PROSER1 in other cell types or tissues with higher Tet expression levels? Is there any mouse gene knockout study for PROSER1?

Answer: Currently no mouse gene knockout model for Proser1 exists. More importantly, we would like to emphasize again that this is the very first study to report a function for PROSER1, thus opening up a platform for future follow up studies. We envision for follow up studies in the future to test TET1 and TET2 protein levels in other PROSER1 KO cell lines such as Proser1 KO mESCs and to establish a Proser1 KO mouse model.

3. Dot blotting data should be complemented by mass spectrometry analysis for a more accurate and reliable determination of genomic content of 5hmC in HEK293 cells.

Answer: We are grateful for the reviewer's insightful comment. Indeed, we have attempted to quantify 5hmC by mass spectrometry but were unable to detect it with DNA purified from HEK293 cells (Fig R2). Our goal for future studies is to utilize genomic DNA from Proser1 KO mESCs which are known to contain higher levels of 5hmC to assess 5hmC in WT and Proser1 KO mESCs.

Sample (100 x)

Figure R2: Mass spectrometry quantification of 5C, 5mC and 5hmC of Degradase-digested genomic DNA purified from HEK293 cells.

References

- Bauer C, Gobel K, Nagaraj N, Colantuoni C, Wang M, Muller U, Kremmer E, Rottach A, Leonhardt H. 2015. Phosphorylation of tet proteins is regulated via o-glcnylation by the o-linked n-acetylglucosamine transferase (ogt). *J Biol Chem.* 290(8):4801-4812. doi:10.1074/jbc.M114.605881
- Hon GC, Song CX, Du T, Jin F, Selvaraj S, Lee AY, Yen CA, Ye Z, Mao SQ, Wang BA, et al. 2014. 5mC oxidation by tet2 modulates enhancer activity and timing of transcriptome reprogramming during differentiation. *Mol Cell.* 56(2):286-297. doi:10.1016/j.molcel.2014.08.026
- Hu L, Li Z, Cheng J, Rao Q, Gong W, Liu M, Shi YG, Zhu J, Wang P, Xu Y. 2013. Crystal structure of tet2-DNA complex: Insight into tet-mediated 5mC oxidation. *Cell.* 155(7):1545-1555. doi:10.1016/j.cell.2013.11.020

- Konzman D, Abramowitz LK, Steenackers A, Mukherjee MM, Na HJ, Hanover JA. 2020. O-glcnac: Regulator of signaling and epigenetics linked to x-linked intellectual disability. *Front Genet.* 11:605263. doi:10.3389/fgene.2020.605263
- Lopez-Moyado IF, Tsagaratou A, Yuita H, Seo H, Delatte B, Heinz S, Benner C, Rao A. 2019. Paradoxical association of tet loss of function with genome-wide DNA hypomethylation. *Proceedings of the National Academy of Sciences of the United States of America.* 116(34):16933-16942. doi:10.1073/pnas.1903059116
- Lu F, Liu Y, Jiang L, Yamaguchi S, Zhang Y. 2014. Role of tet proteins in enhancer activity and telomere elongation. *Genes & development.* 28(19):2103-2119. doi:10.1101/gad.248005.114
- Wang X, Chen X, Sun L, Qian W. 2019. Canonical cytosolic iron-sulfur cluster assembly and non-canonical functions of dre2 in arabidopsis. *PLoS Genet.* 15(4):e1008094. doi:10.1371/journal.pgen.1008094
- Wang X, Li Q, Yuan W, Cao Z, Qi B, Kumar S, Li Y, Qian W. 2016. The cytosolic fe-s cluster assembly component met18 is required for the full enzymatic activity of ros1 in active DNA demethylation. *Sci Rep.* 6:26443. doi:10.1038/srep26443
- Yang X, Qian K. 2017. Protein o-glcnacetylation: Emerging mechanisms and functions. *Nat Rev Mol Cell Biol.* 18(7):452-465. doi:10.1038/nrm.2017.22

September 30, 2021

RE: Life Science Alliance Manuscript #LSA-2021-01228R

Dr. Hans-Martin Herz
St. Jude Children's Research Hospital
262 Danny Thomas Place
Memphis, TN 38105-3678

Dear Dr. Herz,

Thank you for submitting your revised manuscript entitled "PROSER1 mediates TET2 O-GlcNAcylation to regulate DNA demethylation on enhancers and CpG islands". We would be happy to publish your paper in Life Science Alliance pending final revisions necessary to meet our formatting guidelines.

- please add the Twitter handle of your host institute/organization as well as your own or/and one of the authors in our system
- please make sure the author order in your manuscript and our system match
- please upload Figure S8 separately also
- please add sizes next to each blot

A. FINAL FILES:

-- High-resolution figure, supplementary figure and video files uploaded as individual files: See our

detailed guidelines for preparing your production-ready images, <https://www.life-science-alliance.org/authors>

B. MANUSCRIPT ORGANIZATION AND FORMATTING:

Sincerely,

October 4, 2021

RE: Life Science Alliance Manuscript #LSA-2021-01228RR

Dr. Hans-Martin Herz
St. Jude Children's Research Hospital
262 Danny Thomas Place
Memphis, TN 38105-3678

Dear Dr. Herz,

Thank you for submitting your Research Article entitled "PROSER1 mediates TET2 O-GlcNAcylation to regulate DNA demethylation on enhancers and CpG islands". It is a pleasure to let you know that your manuscript is now accepted for publication in Life Science Alliance. Congratulations on this interesting work.

DISTRIBUTION OF MATERIALS:

Again, congratulations on a very nice paper. I hope you found the review process to be constructive and are pleased with how the manuscript was handled editorially. We look forward to future exciting submissions from your lab.

Sincerely,
